# ReactID: Synchronizing Realistic Actions and Identity in Personalized Video Generation

**Wei Li**[1]*, **Yiheng Zhang**[2]†, **Fuchen Long**[2], **Zhaofan Qiu**[2], **Ting Yao**[2], **Xiaoyan Sun**[1,3]†, **Tao Mei**[2]
[1]University of Science and Technology of China,  [2]HiDream.ai Inc.
[3]Institute of Artificial Intelligence, Hefei Comprehensive National Science Center
`weili2023@mail.ustc.edu.cn`, `{yihengzhang.chn, longfuchen}@hidream.ai`,
`{qiuzhaofan, tiyao}@hidream.ai, sunxiaoyan@ustc.edu.cn, tmei@hidream.ai`

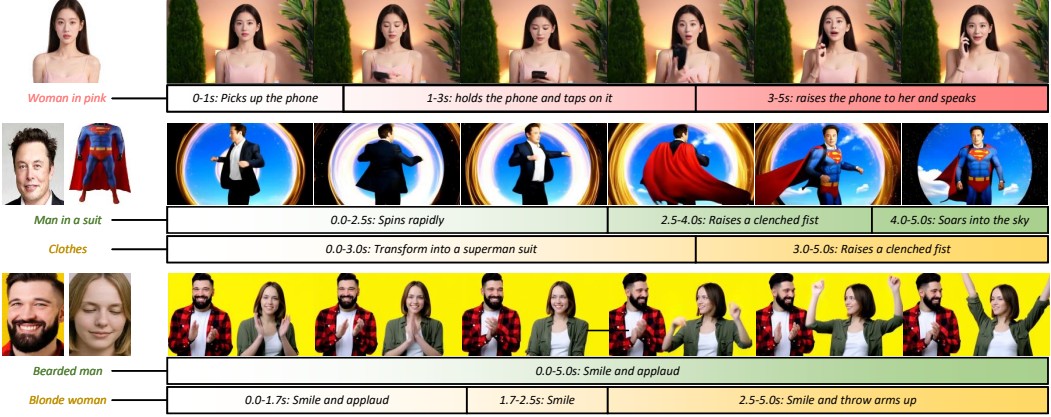

Figure 1: The landscape examples of ReactID for personalized video generation. For each case, ReactID automatically plans a distinct action timeline for each reference subject, generating a final video where each identity performs actions according to their synchronized schedule.

## Abstract

Personalized video generation faces a fundamental trade-off between identity consistency and action realism: overly rigid identity preservation often leads to unnatural motion, while emphasis on action dynamics can compromise subject fidelity. This tension stems from three interrelated challenges: imprecise subject-video alignment, unstable training due to varying sample difficulties, and inadequate modeling of fine-grained actions. To address this, we propose ReactID, a comprehensive framework that harmonizes identity accuracy and motion naturalness through coordinated advances in data, training, and action modeling. First, we construct ReactID-Data, a large-scale dataset annotated with a high-precision pipeline combining vision-based entity label extraction, MLLM-based subject detection, and post-verification to ensure reliable subject-video correspondence. Second, we analyze learning difficulty along dimensions such as subject size, appearance similarity, and sampling strategy, and devise a progressive training curriculum that evolves from easy to hard samples, ensuring stable convergence while avoiding identity overfitting and copy-paste artifacts. Third, ReactID introduces a novel timeline-based conditioning mechanism that supplements monolithic text prompts with structured multi-action sequences. Each sub-action is annotated with precise timestamps and descriptions, and integrated into the diffusion model via two novel components: subject-aware cross-attention module to bind sub-action to the specific subject of interest and temporally-adaptive RoPE to embed the rescaled temporal coordinates invariant to action duration. Experiments show that ReactID achieves state-of-the-art performance in both identity preservation and action realism, effectively balancing the two objectives.

---

*This work was performed at HiDream.ai.
†Corresponding authors.

Figure 2: An illustration of three key components in ReactID to improve personalized video generation from the perspectives of data construction, training paradigm and timeline formulation.

# 1 INTRODUCTION

The generation of high-fidelity video content has emerged as a pivotal task in computer vision and graphics, with profound applications in content creation, virtual reality, and personalized media. A particularly challenging and impactful subset of this field is Personalized Video Generation, which aims to create video sequences of a specific subject (e.g., a person or object) performing a desired action while maintaining the identity of subject across frames. The core challenge in this task lies in achieving an optimal balance between two competing objectives: *preserving the identity of subject with high fidelity* and *ensuring the realism and naturalness of the generated actions*.

Current state-of-the-art approaches (Chen et al., 2025b; Huang et al., 2025; Fei et al., 2025; Liu et al., 2025b; Hu et al., 2025a; Jiang et al., 2025), typically built upon diffusion-based architectures, attempt to approach personalized video generation by incorporating reference identity information through cross-attention mechanisms or adapter modules. Despite the promising results, these methods consistently fall short of achieving the ideal balance between identity consistency and action realism. We speculate that this limitation stems from three fundamental bottlenecks in the generation pipeline, each contributing to the imbalance in different ways:

(1) **Inaccurate identity preservation** resulting from noisy subject-video correspondences directly undermines the identity aspect of the balance. Conventional annotation pipelines often produce incomplete subjects, misaligned bounding boxes, and erroneous identity associations. When models learn from such imperfect supervision, they develop unreliable identity representations, leading to either identity drift or visual artifacts in generated videos.

(2) **Unstable convergence patterns** caused by varying learning difficulties across samples disrupt the training process itself. Easy samples (with large, clear, and temporally consistent entities) encourage the model to adopt a "copy-paste" strategy, preserving identity at the expense of natural motion. Conversely, hard samples (with small scale, or appearance variations), though slowing convergence and complicating identity alignment, serve a critical function: they compel the model to maintain and leverage its pre-trained generative priors, thereby enhancing realistic action synthesis and mitigating catastrophic forgetting of motion dynamics.

(3) **Compromised action naturalness** often results from coarse-grained action modeling that fail to capture temporal dynamics. Without precise temporal guidance, models tend to prioritize identity features—which are more explicitly supervised—over motion patterns, resulting in stiff, unnatural movements that diminish the overall realism of generated videos.

In this paper, we propose **ReactID**, a comprehensive framework designed to synchronize realistic action generation with faithful identity preservation through synergistic improvements across data curation, training strategy, and action modeling, as illustrated in Figure 2. Specifically, we first introduce ReactID-Data, a large-scale dataset with unprecedented annotation quality, enabled by a novel automated pipeline that leverages vision-based entity word extraction, Multi-modality Large Language Models (MLLM) based detection and additional post-verification to ensure reliable subject-video correspondences. Then, we develop a difficulty-aware progressive learning strategy that explicitly manages sample difficulty based on subject size, appearance similarity, and sampling strategy. Our curriculum learning approach ensures stable convergence while preventing both overfitting to easy samples and underfitting on challenging ones. To enable the modeling of complex, multi-action sequences, ReactID exploits a structured timeline-based conditioning mechanism complement to the conventional use of a single, monolithic text prompt for video generation. This timeline is composed of multiple fine-grained sub-actions, each annotated with precise start and end times alongside a detailed textual description of the specific motion. To effectively inject this rich

spatio-temporal information into a diffusion-based generator, we introduce two key novel components: a subject-aware cross-attention module and a temporally-adaptive RoPE mechanism. The former embeds the rescaled temporal coordinates of the sub-actions, enhancing the compatibility and robustness to actions of varying lengths. The latter explicitly binds the descriptive sub-action cues to the specific subject of interest within the scene, ensuring that the generated motions are accurately associated with the correct subject. This synergistic integration enables ReactID to achieve unprecedented control over personalized video generation, seamlessly synchronizing realistic actions with subject identity.

The main contributions of this work are summarized as follows. We propose ReactID, an innovative framework that effectively harmonizes identity preservation and action realism in personalized video generation. To support this framework, we introduce a large-scale, high-precision dataset (ReactID-Data), a progressive training strategy that transitions from easy to difficult samples, and a novel modeling architecture capable of representing complex multi-action sequences. Through extensive experiments, ReactID demonstrates state-of-the-art performance in both identity consistency and motion naturalness. Moreover, we believe the release of ReactID-Data with its temporally-precise multi-action annotations will significantly facilitate future research in personalized video generation.

## 2 RELATED WORK

**Video Diffusion Models.** Recently, text-to-video (T2V) generation has witnessed remarkable progress, largely driven by advances in diffusion models. Early attempts (Blattmann et al., 2023; Guo et al., 2024; Zhang et al., 2024; 2025c) built on UNet (Ronneberger et al., 2015) architecture are restricted to synthesize short clips with limited spatial and temporal resolution. Facing the unprecedented capability demonstrated by Sora (Brooks et al., 2024) in generating high-quality, minute-long videos, transformer-based diffusion models (Peebles & Xie, 2023) are increasingly introduced as the mainstream, replacing UNet backbones and enabling scalability. Among these, MMDiT (Esser et al., 2024), a dual-stream DiT architecture introduced by Stable Diffusion 3, is subsequently adopted in open-source video diffusion projects such as CogVideoX (Yang et al., 2025) and HunyuanVideo (Kong et al., 2024). More recently, Wan et al. (2025) and Cai et al. (2025) propose novel video DiT structures, and achieve remarkable performance in realistic video generation.

**Personalized Video Generation.** The field of personalized video generation has witnessed a rapid evolution from early UNet-based approaches (Chefer et al., 2024; He et al., 2024b; Ma et al., 2024; Wu et al., 2024; Wang et al., 2024b; Xu et al., 2025b; Chen et al., 2024; Li et al., 2025b; Wu et al., 2025) to advanced Diffusion Transformer (DiT) architectures (Yuan et al., 2025b; Zhang et al., 2025b; Wei et al., 2025a). Initial efforts on UNet were often confined to simple single-subject scenarios (He et al., 2024b; Ma et al., 2024; He et al., 2024b; Jiang et al., 2024; Wei et al., 2024b), struggling with limited motion and the need for extensive test-time tuning (Wei et al., 2024a; Wu et al., 2024). DreamVideo (Wei et al., 2024a) first decouples the learning for subject and motion. CustomCrafter (Wu et al., 2025) finetunes self-attention layers to enable subject customization without disrupting the inherent motion modeling. With the paradigm shift to DiT frameworks, recent works have substantially advanced visual fidelity and tuning-free capabilities. Owing to the scalable nature of DiT, the scope of personalization has been broadened to encompass complex multi-subject and open-domain settings (Huang et al., 2025; Chen et al., 2025b; Liu et al., 2025b; Jiang et al., 2025; Fei et al., 2025; Hu et al., 2025a; Deng et al., 2025; Hu et al., 2025b; Xu et al., 2025a). Moreover, customization learning are not limited to visual appearance and identity, but also extended to relations and interactions. Pioneering this direction, ReVersion (Huang et al., 2024) proposes a diffusion-based relation inversion framework to capture specific relation from images, which is further employed for image generation. DreamRelation (Wei et al., 2025b) further extends such type of relation modeling into video synthesis. Despite these advancements, a key challenge persists: the inherent trade-off between subject identity preservation and accurate motion generation often results in "copy-paste" artifacts and unnatural motion. Compounding this issue is the scarcity of high-quality training data, which remains a primary bottleneck. This underscores the urgent need for datasets with high action precision, video temporal consistency, and reference image diversity.

**In summary**, our work mainly focuses on preserving subject identity with high fidelity and ensuring natural actions in personalized video generation. The proposal of ReactID contributes by studying not only how to conduct progressive subject-to-video learning for accurate identity preservation while alleviating "copy-paste" artifacts, but also how to integrate the timeline information into video diffusion to achieve precise action realism.

## 3    METHOD

### 3.1    REACTID-DATA: A DATASET FOR PERSONALIZED VIDEO GENERATION

**Data Preparation.** We initialize a pool of 20 million videos from publicly available video collections such as HD-VG-130M (Wang et al., 2025) and OpenHumanVid (Li et al., 2025a). Then we employ a pipeline to process raw videos, which involves key stages such as scene detection, video transcoding, text removal, quality assessment and filtering, and subject-centric captioning. The implementation details for this workflow are elaborated in the supplementary materials.

**Entity Extraction.** A typical method to identify entities from video captions is to extract nouns via a Named Entity Recognition (NER) model (e.g., SpaCy (Honnibal & Montani, 2017)) with a pre-defined taxonomy. However, the limited taxonomy results in coarse-grained entities that cannot distinguish between different instances, especially for living subjects. For instance, two distinct individuals might both be labeled simply as "a person." To mitigate this issue, we process living and non-living subjects differently. We first construct a 1200-term taxonomy by combining nouns from captions with labels from image datasets (e.g., Deng et al. (2009), Caesar et al. (2018), and Shao et al. (2019)), and then divide the terms into living and non-living categories. For each video-caption pair, an NER model guided by our taxonomy is employed to extract all entities from the caption. Entities identified as non-living subjects are directly retained as the final result, whereas those corresponding to living subjects undergo further processing. For these cases, a vision-language model analyzes the video content to generate a fine-grained, descriptive entity label for each living subject. As a result, we obtain specific entity labels for living subjects like "the person in red" and "the person sitting on the bench", rather than generic ones such as "a person" and "another person", enabling more precise entity referencing, which is curial for the following processing.

**Subject Detection and Segmentation.** Next, extracted entities are grounded to specific spatio-temporal regions within the video via subject detection and segmentation, thus forming comprehensive entity annotations. The primary challenge in this process lies in distinguishing between visually/semantically similar subjects and ensuring their correct association with the corresponding entities. To achieve this, an MLLM-based detector, Florence-2 (Xiao et al., 2024), is employed to locate the bounding boxes of the given entity labels, which are further verified by examining the cross-modal distance in the SigLIP (Zhai et al., 2023) feature space. Conditioning on the bounding boxes, we obtain the segmentation masks via the SAM (Ravi et al., 2024) model for each entity. Moreover, to better support face-centric identity-preserving generation tasks, we additionally extract face bounding boxes and masks for "human" entities utilizing the InsightFace Toolbox (Deng et al., 2019) and the SAM model, respectively.

### 3.2    DIFFICULTY-AWARE CURRICULUM LEARNING

A training sample consists of one or more reference images $I_{ref}$ cropped from a subject's bounding box, a video clip in which the subject appears $V_{gt}$, and a video caption $text$. Easy samples, where the subject in the video is large and highly similar to the reference, can encourage a "copy-paste" shortcut that harms generalization. In contrast, harder samples with small or varied subjects in videos force the model to learn the desired skill of fusing identity in $I_{ref}$ with action and context provided in $text$, but they also present a convergence challenge. To balance the easy and hard samples and effectively leverage our ReactID-Data, we introduce an easy-to-hard curriculum learning strategy that ensures stable convergence while avoiding identity overfitting and "copy-paste" artifacts. Technically, we formalize this difficulty-aware learning by quantifying sample difficulty using three cues: subject size, appearance similarity, and sampling strategy.

**Subject size** refers to the spatial proportion of the subject within the video frames. We define the difficulty score $\mathcal{D}_{sub}$ for each sample. Let the video $V_{gt}$ consist of $N$ frames, each with a resolution of $H \times W$. For the $n$-th frame, a binary segmentation masks $M_n$ is used, where $M_n(i,j) = 1$ indicates that the pixel belongs to the subject in the reference image. The difficulty is calculated as:

$$\mathcal{D}_{sub} = 1 - (NHW)^{-1} \sum_{n=1}^{N} \sum_{i=1}^{H} \sum_{j=1}^{W} \mathbf{1}_{\{M_n(i,j)=1\}} \ , \tag{1}$$

where $\mathbf{1}_{\{\cdot\}}$ is the indicator function. This formulation ensures that a smaller subject yields a higher difficulty score $\mathcal{D}_{sub}$, which approaches 1.

**Appearance similarity** quantifies the identity alignment between the reference image $I_{ref}$ and the subject's appearance in the video. We measure this by first detecting all $N$ subject regions $\{P_i\}_1^N$ in the video and then calculating the average cosine similarity between the feature embeddings of $I_{ref}$ and each $P_i$. The difficulty score $\mathcal{D}_{app}$ is defined as:

$$\mathcal{D}_{app} = 1 - N^{-1} \sum\nolimits_{i=1}^{N} \left[ \mathcal{F}(I_{ref}) \cdot \mathcal{F}(P_i) \right] / \left[ \|\mathcal{F}(I_{ref})\| \cdot \|\mathcal{F}(P_i)\| \right] \quad , \tag{2}$$

where $\mathcal{F}(\cdot)$ is DINOv2 (Oquab et al., 2023) for general subjects and ArcFace (Deng et al., 2019) for faces. This formulation ensures that lower similarity results in a higher score.

**Sampling strategy** evaluates difficulty based on whether the reference image and video originate from the same clip. We define two strategies: *intra-clip sampling* (easy, $\mathcal{D}_{sam} = 0$), and *inter-clip sampling* (hard, $\mathcal{D}_{sam} = 1$). In intra-clip sampling, the reference $I_{ref}$ is taken directly from the video $V_{gt}$, ensuring high visual consistency (lighting, background). Conversely, inter-clip sampling pairs the video with a visually similar reference from a different clip, forcing the model to handle greater variation and thus posing a more challenging learning task.

Finally, to obtain a single, comprehensive difficulty metric for each training sample, we compute a weighted sum of the scores from the three cues: $\mathcal{D}_{overall} = \lambda_{sub}\mathcal{D}_{sub} + \lambda_{app}\mathcal{D}_{app} + \lambda_{sam}\mathcal{D}_{sam}$, where the $\lambda_{sub}$, $\lambda_{app}$, and $\lambda_{sam}$ are trade-off parameters that are determined empirically. During model training, we implement our curriculum by imposing a difficulty threshold $\tau$ and only samples where $\mathcal{D}_{overall} \leq \tau$ are used for training. As the training progresses, the threshold $\tau$ is increased in a multi-step schedule, progressively introducing more challenging examples for training. We pre-sort the training data according to the difficulty score $\mathcal{D}_{overall}$ and derive the difficulty threshold $\tau$ from the statistical quartiles of the score distribution. Following the ratio of 4:2:1:1, we partition the total training steps into four phases. At the transition of each phase, $\tau$ is stepped up to the next quartile. After each update point, the threshold remains fixed until the next scheduled change.

### 3.3 TEMPORAL-AWARE ACTION MODELING

#### 3.3.1 TIMELINE ANNOTATION CONSTRUCTION

Modern text-to-video and subject-to-video datasets normally construct the training data as the pair of text-video or the triplet of text-video-reference, and the granular timeline information of actions is scarcely provided. The lack of temporal annotation could limit the model capacity to characterize sophisticated time-related action generation (Zhang et al., 2025a; Qiu et al., 2025). In the scenario of personalized video generation, exploiting additional timeline information in training can further precisely regulate the fine-grained sub-action generation, facilitating the synthesis of more natural motion dynamic. To obtain such time-aware information for training, we sample a subset with 1.2M subject-to-video pairs from the ReactID-Data, and further construct the detailed timeline annotations for each video.

Specifically, we devise an ensemble approach that combines the vision language model (VLM), i.e., Qwen2.5-VL-72B (Bai et al., 2025), with two visual temporal grounding models UniMD (Zeng et al., 2024) and TFVTG (Zheng et al., 2024) for timeline annotation construction. The VLM is responsible for initial temporal localization of events, and the two grounding models provide additional two sets of timestamp based on the initial captions generated by Qwen. Finally, we exploit another VLM, i.e., InterVideo2 (Wang et al., 2024a) as a scoring model to select the best candidates from the three annotations. Please refer Appendix B.2 for more details about the timeline annotation.

#### 3.3.2 TEMPORAL-AWARE SUBJECT INTEGRATION

The overview of our ReactID is illustrated in Figure 3. To handle the subject timeline integration in diffusion, we introduce a novel subject-synchronized module which consists of two specific designs, i.e, subject-aware cross-attention and temporally-adaptive RoPE, in DiT blocks.

**Subject-Aware Cross-Attention.** For the subject-synchronized module as depicted in Figure 3, we exploit an attention-based mask predictor to estimate the mask for each subject, and further employ the masks to associate subject-related video latents and timeline prompts in cross-attention. In detail, we first compute the cross-attention map between reference image tokens and video tokens. Then, we feed the attention map into MLP for mask prediction for each subject, which is supervised by the ground-truth subject mask using a focal loss. To enhance the accuracy of subject mask prediction,

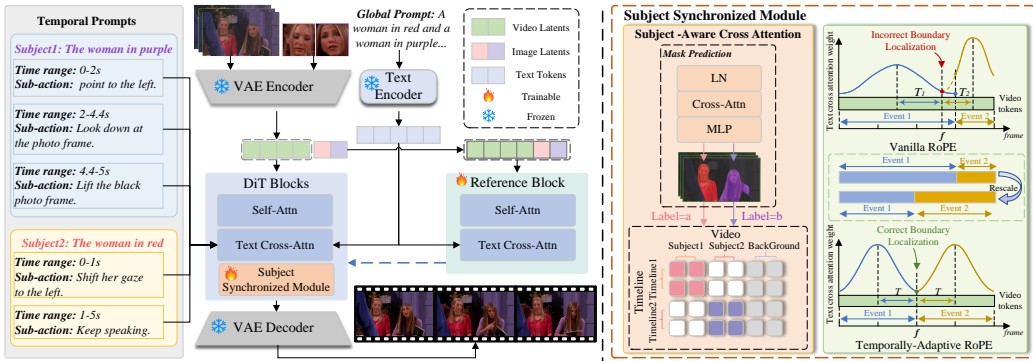

Figure 3: An overview of ReactID framework for personalized video generation.

we average all results from last five DiT blocks to obtain final subject masks. Next, we implement a label binding mechanism to establish explicit correspondence between subject-related video tokens and their sub-action timeline prompt in cross-attention. Specifically, we assigned a unique numerical label for subject-related video tokens located in the mask, and the corresponding timeline prompt receiving the same label. For instance, video tokens of "woman in red" receives label $\alpha$, while video tokens of "woman in purple" is assigned label $\beta$. Video tokens falling outside all subject masks are classified as background and assigned a distinct label that differs from any subject label. Given the assigned labels, a label-dependent phase modulation is proposed to facilitate video tokens to primarily focus on their corresponding subject's timeline while maintaining global interaction in cross-attention. The phase adjustments on both the query $q_i$ (i.e, video tokens) and the key $k_j$ (i.e., timeline prompt embeddings of each sub-action) are formulated as follows:

$$\tilde{q}_i = q_i e^{l_i \theta_0}, \quad \tilde{k}_j = k_j e^{l_j \theta_0}, \tag{3}$$

where $l_i$ and $l_j$ denote the assigned label value, and $\theta_0$ is the base angle. Such phase adjustment ensures that the attention bias $\tilde{q}_i^\top \tilde{k}_j$ is maximized when label $l_i$ and $l_j$ match.

The subject-aware cross-attention guarantees that the video tokens with matched subject labels precisely attend to the timeline prompts at spatial level. The tokens with matching labels can be well-aligned, and those with mismatched labels are softly separated, ensuring the textual context embedded in the timeline annotations are correctly routed to the corresponding subject.

**Temporally-Adaptive RoPE.** In addition to spatial-level attention design to enhance subject and sub-action association, the precise timeline-aligned personalized video generation also necessitates the subject-action correlation modeling in temporal dimension. Vanilla temporal RoPE (Rotary Position Embedding) in attention exploits the absolute frame indices to construct the embedding, implicitly assuming that all sub-actions have the same duration. Nevertheless, the durations of sub-actions typically vary, and directly applying vanilla RoPE will lead to attention bias misalignment near the transition boundaries between two adjacent sub-actions. As demonstrated in Figure 3, the distance between the frame $f$ and the midpoint of long sub-action is more than that between $f$ and the midpoint of the subsequent short sub-action. Thus, $f$ will perceive more attention bias to the midpoint of the short sub-action than the long sub-action when using vanilla RoPE. The allocation of attention bias is not reasonable since $f$ belongs to the long sub-action, and this will lead to inaccurate temporal modeling and abrupt transitions. To alleviate this, we propose temporally-adaptive RoPE that rescales the temporal axis when constructing the temporal RoPE, which is invariant to the duration of various sub-action. Taking the frame index $f$ that locates in $n$-th time interval $\left[ f_n^{\text{start}}, f_n^{\text{end}} \right]$ as an example, the rescaled index $f'$ is formulated as:

$$f' = (f - f_n^{\text{start}})/(f_n^{\text{end}} - f_n^{\text{start}}) \cdot T + (n-1) \cdot T, \quad \text{s.t. } f_n^{\text{start}} \le f \le f_n^{\text{end}}, \tag{4}$$

where $T$ is a pre-defined constant that rescales each sub-action into a uniform temporal length.

**Inference with Single Prompt.** In addition to the personalized video generation using the structured timeline inputs, our ReactID can be readily extended for video generation given a single natural prompt. To enable this feature, we leverage LLMs to function as an automated temporal planner, which converts the single prompt into the timeline formats. The input prompt is first expanded into a detailed global description, and then decomposed into a sequence of distinct event captions, assigned with a plausible time interval for each sub-action. The obtained results are finally formed as structured timeline for ReactID to generate videos with realistic actions and identity preservation.

Table 1: Quantitative comparison of different approaches on the personalized video generation task across metrics: Aesthetics, Motion Smoothness, Motion Amplitude, Face Similarity, GmeScore, NexusScore, and NaturalScore. The best-performing result is highlighted in **bold**, and the second-best is underlined.

| Method | Aes.↑ | M. Smo.↑ | M. Amp.↑ | FaceSim↑ | Gme.↑ | Nexus.↑ | Natural.↑ | Total.↑ |
|---|---|---|---|---|---|---|---|---|
| *OpenS2V-Eval: Single Domain* | | | | | | | | |
| VACE-P1.3B | 48.93% | 95.68% | 11.91% | 18.04% | 70.78% | 36.24% | 66.85% | 49.20% |
| VACE-1.3B | 49.41% | 95.42% | 22.51% | 22.37% | 70.87% | 38.34% | 68.33% | 51.13% |
| Phantom-1.3B | 49.00% | 93.70% | 16.38% | **44.03%** | 69.54% | 37.72% | 66.76% | 54.50% |
| **ReactID** | **49.79%** | **96.25%** | **38.21%** | 40.80% | **71.16%** | **39.85%** | **70.29%** | **56.04%** |
| *OpenS2V-Eval: Human Domain* | | | | | | | | |
| ID-Animator | 42.03% | 94.69% | 33.54% | 31.56% | 52.91% | – | 56.11% | 49.75% |
| ConsisID | 41.77% | 79.83% | 37.99% | 43.19% | 72.03% | – | 55.83% | 54.19% |
| EchoVideo | 39.93% | 77.96% | 35.58% | 48.65% | 68.40% | – | 62.22% | 56.36% |
| FantasyID | 45.60% | 85.44% | 23.41% | 32.48% | 72.68% | – | 62.36% | 54.33% |
| VACE-P1.3B | 51.91% | 95.80% | 8.78% | 19.98% | 73.27% | – | 65.83% | 53.97% |
| VACE-1.3B | **53.18%** | 95.84% | 16.87% | 22.29% | 73.61% | – | 65.28% | 54.90% |
| Phantom-1.3B | 50.80% | 92.02% | 14.09% | 46.29% | 72.17% | – | 65.83% | 60.00% |
| Concat-ID-Wan-AdaLN | 43.13% | 85.86% | 17.19% | 50.05% | 71.90% | – | 68.47% | 59.85% |
| **ReactID** | 52.02% | **96.46%** | **40.79%** | 44.08% | **73.96%** | – | 69.31% | **62.17%** |
| *ReactID-Eval-SEQ* | | | | | | | | |
| VACE-P1.3B | 46.59% | 90.07% | 15.81% | 13.52% | 69.83% | 34.22% | 60.68% | 45.66% |
| VACE-1.3B | 47.92% | 92.73% | 23.65% | 19.48% | 70.02% | 35.18% | 64.78% | 48.58% |
| Phantom-1.3B | 45.68% | 92.16% | 15.38% | 37.41% | 67.35% | **38.69%** | 62.30% | 51.40% |
| **ReactID** | **49.11%** | **94.58%** | **39.46%** | **38.20%** | **71.23%** | 37.13% | **68.69%** | **54.42%** |

# 4 EXPERIMENTS

## 4.1 EXPERIMENTAL SETTINGS

**Datasets and Metrics.** OpenS2V-Eval (Yuan et al., 2025a) is a fine-grained benchmark that contains 180 subject-text pairs focusing on the model's ability to generate subject-consistent videos with natural subject appearance and identity fidelity. Moreover, to better assess the generation of complex, multi-action sequences, we introduce an evaluation set, ReactID-Eval-SEQ. It comprises 120 subject-text pairs, encompassing 40 human identities and 20 animals. The text prompt in each pair specifies sequential actions for the subject. Our evaluation methodology adheres to the OpenS2V-Nexus (Yuan et al., 2025a), employing seven scores to assess the generated videos: Aesthetics, Motion Smoothness, Motion Amplitude, Face Similarity, GmeScore, NexusScore, and NaturalScore. In line with the OpenS2V-Nexus protocol, all scores are normalized and then aggregated into a final TotalScore, with both processes following the procedures specified therein.

**Implementation Details.** We employ Wan2.1-T2V-1.3B (Wan et al., 2025) as our foundational model. Our ReactID model is trained for 10k steps with a global batch size of 32, utilizing the AdamW optimizer with a learning rate of $1 \times 10^{-5}$ and 500 warmup steps. All training is conducted on 8 NVIDIA A100 GPUs, and consumes total $1,464$ GPU hours. The averaged time of each optimization step is 65.8 seconds. Regarding the curriculum learning, we initialize the difficulty threshold $\tau$ at $0.53$ and update it to $0.67$, $1.44$, and $1.84$ at training steps 5k, 7.5k, and 8.75k, respectively. The ratio of inter-clip references within the total training samples is dynamically modulated by $\tau$, adjusted to 0%, 0%, 11% and 33% at the corresponding update points. Since our proposal is insensitive to $\lambda_{sub}$, $\lambda_{app}$, and $\lambda_{sam}$, we empirically set them to $0.5$, $1$, $1$, respectively, according to the relative importance of each dimension. For inference, we use 50 denoising steps and a classifier-free guidance (CFG) scale of 5.0. Generating a 5-second video requires about $316$ seconds. An LLM (Achiam et al., 2023) is employed to convert prompts into a timeline format when necessary.

## 4.2 EVALUATION OF REACTID

**Quantitative Results.** In Table 1, we first present an analysis of our proposed ReactID in comparison with several open-source approaches on the OpenS2V-Eval across two scenarios: the Single Domain, which consists of test samples each containing a single subject drawn from various categories, and the Human Domain, which is composed solely of subjects from the 'person' category. All the mentioned baselines utilize foundational models with complexities comparable to ours, providing a fair basis for evaluation. In general, ReactID outperforms baselines across most metrics and achieves the highest overall scores in both domains. In the Single Domain, ReactID achieves superior performance across most metrics, including NexusScore, GmeScore, NaturalScore, and To-

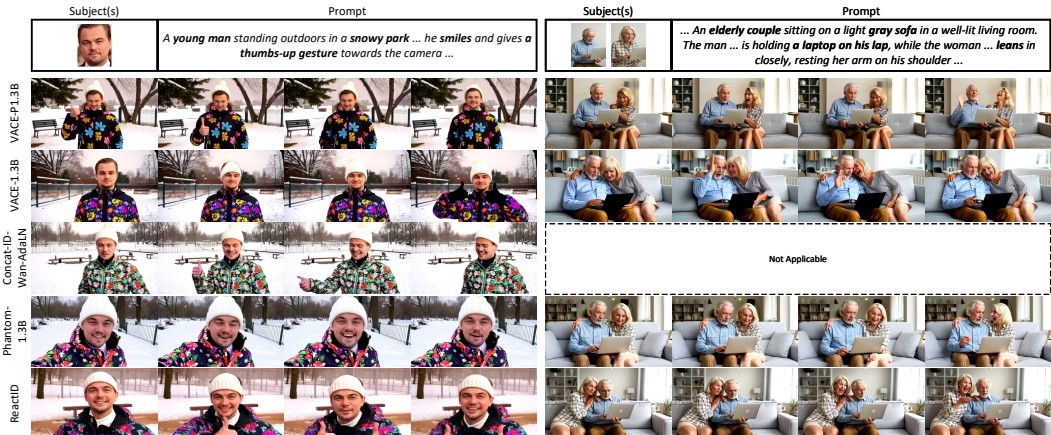

Figure 4: Qualitative comparisons of videos produced by different approaches on the OpenS2V-Eval dataset.

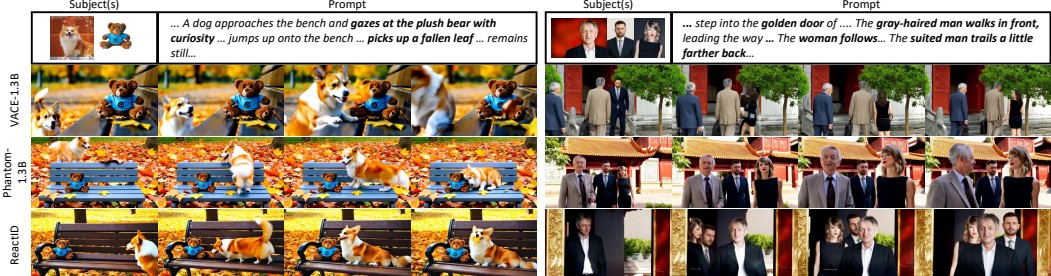

Figure 5: Comparisons of videos produced by different approaches on the ReactID-Eval-SEQ dataset.

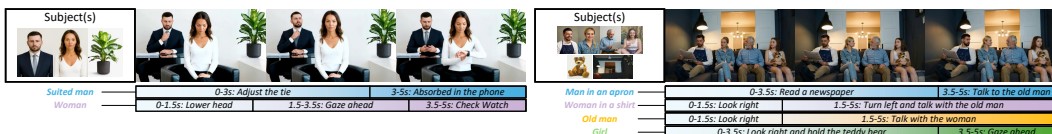

Figure 6: Examples of video generation conditioned on multiple subjects and their respective timelines.

talScore. This highlights its capability to effectively harmonize subject consistency with action realism. However, it scores marginally lower than Phantom-1.3B on FaceSim. We attribute this deficit to ReactID generating more dynamic and natural movements (higher motion amplitude/smoothness), which may introduce motion blur, thereby reducing facial similarity to the reference image. In the Human Domain, ReactID achieves a TotalScore of 62.17%, surpassing the strongest of the seven baselines, Phantom-1.3B, by 2.17%. It also secures the top position in four out of the six available metrics (NexusScore is not applicable in this domain). Similar to the results in the Single Domain, ReactID achieve a competitive FaceSim score of 44.08% with a high Motion Amplitude of 40.79%. To further demonstrate ReactID's advantages in generating videos with sequential actions, we conducted experiments on the ReactID-Eval-SEQ. We compared our model with three baseline approaches supporting both human and animal subjects. The results show that ReactID outperforms all baselines across most metrics. Notably, its TotalScore is 3.02% higher than the best competitor, which highlights ReactID's capability for sequential action generation.

**Qualitative Results.** We proceed with a visual examination of the generation quality, comparing ReactID against four baselines (VACE-P1.3B, VACE-1.3B, Concat-ID-Wan-AdaLN, and Phantom-1.3B) on two input subject-text pairs. As illustrated in Figure 4, while all approaches successfully generate videos featuring the subjects from the reference images, our model produces the most plausible visual quality and naturalistic motions. Taking the left case as an example, ReactID adeptly generates a man in a snowy park giving a thumbs-up gesture. In contrast, the baseline VACE-P1.3B and VACE-1.3B struggles to preserve the subject's key visual attributes from the reference image, leading to a degradation in subject consistency. While Phantom-1.3B effectively replicate the subject's appearance, it fails to follow the text prompt and its video quality is compromised by "copy-paste" artifacts. These artifacts manifest as subjects appearing static or being unnaturally duplicated across frames, which results in visually implausible motion sequences.

Table 2: Comparisons between Re-
actID and four baselines with re-
gard to human preference.

| Method | SC↑ | AN↑ | TVA↑ | OVQ↑ |
|---|---|---|---|---|
| ConsisID | 1.97 | 1.66 | 2.13 | 1.84 |
| ConcatID-WA | 3.06 | 2.10 | 2.91 | 2.74 |
| Phantom-1.3B | 3.46 | 2.97 | 2.88 | 3.25 |
| VACE-1.3B | 3.23 | 3.20 | 3.08 | 3.37 |
| **ReactID** | **3.90** | **4.00** | **3.47** | **3.42** |

Table 3: Results on label binding in
Subject-Aware Cross-Attention.

| Method | FaceSim↑ | Gme.↑ | Natural.↑ |
|---|---|---|---|
| *Label Strategy* | | | |
| Adaptive | 48.42% | 70.16% | 66.24% |
| Uniform | **49.11%** | **71.23%** | **68.69%** |
| *Label Value Selection: $(\alpha, \beta)$* | | | |
| $(0, 1)$ | **49.34%** | 70.08% | 68.51% |
| $(2, 20)$ | 49.11% | **71.23%** | **68.69%** |

Table 4: Ablation results on Tem-
porally Adaptive RoPE in terms of
strategy and rescale length.

| Method | CLIP-T↑ | TVA↑ | TC↑ |
|---|---|---|---|
| Attention Mask | 0.241 | 2.62 | 2.25 |
| Vanilla RoPE | 0.242 | 2.67 | 2.35 |
| TARoPE $T$=8 | 0.246 | 2.70 | 2.36 |
| TARoPE $T$=4 | 0.247 | 2.69 | 2.35 |
| TARoPE $T$=2 | **0.247** | **2.71** | **2.38** |

Table 5: Ablation study of the difficulty-aware curriculum learning in ReactID.

| Method | Aes.↑ | M. Smo.↑ | M. Amp.↑ | FaceSim↑ | Gme.↑ | Nexus.↑ | Natural.↑ | Total.↑ |
|---|---|---|---|---|---|---|---|---|
| Full Data Training | 48.19% | 92.64% | 37.22% | 34.89% | 69.16% | 33.10% | 65.13% | 51.54% |
| Random Expansion | 48.46% | 92.56% | 37.46% | 34.74% | 69.11% | 33.24% | 64.36% | 51.39% |
| Curriculum w/o $D_{sub}$ | 48.43% | 93.68% | 35.27% | 38.03% | 70.68% | 34.33% | 68.01% | 53.35% |
| Curriculum w/o $D_{app}$ | 48.96% | 94.07% | 38.84% | 37.18% | 71.16% | 35.89% | 68.14% | 53.76% |
| Curriculum w/o $D_{sam}$ | 48.57% | 93.26% | 37.87% | 35.76% | 70.41% | 35.06% | 66.43% | 52.68% |
| **Full Curriculum** | **49.11%** | **94.58%** | **39.46%** | **38.20%** | **71.23%** | **37.13%** | **68.69%** | **54.42%** |

**Evaluation of ReactID-Data.** A comparative analysis of our ReactID-Data against OpenS2V-5M (Yuan et al., 2025a), which includes evaluations of data quality, human preference, and model training impact, is detailed in the supplementary materials.

**User Study.** We further conducted a user study to evaluate whether videos produced by ReactID better align with human preferences compared to those from four baseline methods: ConsisID, Concat-ID-Wan-AdaLN (ConcatID-WA), Phantom-1.3B, and VACE-1.3B. For this study, we randomly sampled 50 subject-text pairs from OpenS2v-Eval to serve as test cases. We recruited 25 evaluators (17 males, 8 females) with diverse educational backgrounds and ages. Each participant was asked to rate the generated videos on a 5-point Likert scale across four critical dimensions: (1) subject consistency (SC), (2) action naturalness (AN), (3) text-video alignment (TVA), and (4) overall visual quality (OVQ). To ensure a reliable assessment, the final score for each dimension was computed by averaging ratings across all participants and test cases. As presented in Table 2, ReactID demonstrates consistent superiority over all baselines across all four evaluated dimensions. This study thus validates the effectiveness of ReactID from a human perceptual standpoint.

## 4.3 ABLATION STUDY ON REACTID

In this section, we perform a series of ablation studies to delve into the design of ReactID for personalized video generation. Note that all results here are evaluated on the ReactID-Eval-SEQ.

**Difficulty-aware Curriculum Learning.** We first delve into the effectiveness of our difficulty-aware curriculum learning for personalized video generation, and report the performances in Table 5. When progressively expanding the training set without using any difficulty guidance (i.e., **Random Expansion**), the model achieves 51.39% TotalScore, which is lower than that of training with all data (i.e, 51.54% achieved by **Full Data Training**). To ablate each component of difficulty-aware curriculum learning, we conduct experiments by discarding any one of the three difficulty cues, and the performances are superior to the baseline of no curriculum. By further exploiting all the three cues in curriculum learning to avoid both overfitting to easy samples and underfitting on challenging ones, ReactID with **Full Curriculum** learning shows the best 54.42% TotalScore.

**Subject-Aware Cross-Attention.** Then, we investigate different technical choices of label binding strategies in subject-aware cross-attention of subject synchronized module. Table 3 details the performances of two runs, i.e., adaptive labeling and our uniform labeling. **Adaptive labeling** assigns varying label values based on predicted mask probabilities. The pixels with higher probability receive larger values, while pixels with lower probability are assigned smaller ones. Instead, **Uniform labeling** assigns a fixed label value to all pixels within each predicted subject mask. Counterintuitively, the uniform labeling strategy outperforms the adaptive counterpart across all metrics. We speculate that the results may be caused by the misalignment between the attended region derived from mask probabilities and the region essential for personalized video generation. Compared to potentially erroneous adaptive labeling, uniformly applying the same label to all tokens in the mask can be more robust to attain better identity preservation. For the uniform labeling, we further explore the influences of the labeling value selection. As indicated by the results, exploiting more distinct label values (i.e., $\alpha = 2$ and $\beta = 20$) for different subjects empirically achieves better performances.

Table 6: Ablation study of designs w.r.t. timeline.

| Model | Data | TotalScore |
|---|---|---|
| Base | OpenS2V-5M | 50.37% |
| Base | ReactID⁻ | 52.14% |
| Base | ReactID | 52.21% |
| ReactID | ReactID | 54.42% |

Table 7: Annotation quality of timeline and mask.

| Timeline | | Mask | |
|---|---|---|---|
| Method | F1 | Method | P. Rate |
| TFVTG | 0.57 | LISA | 15% |
| InternVL | 0.63 | SAM | 31% |
| Qwen | 0.71 | ReactID | 54% |
| ReactID | 0.78 | | |

Table 8: User preference of timelines from Expert/LLM.

| Dims | LLM | Expert |
|---|---|---|
| AO | 46% | 54% |
| LC | 48% | 52% |
| MN | 53% | 47% |

Table 9: Human Distinguishability of timelines from Expert/LLM.

| PD / GT | LLM | Expert |
|---|---|---|
| LLM | 52.8% | 47.2% |
| Expert | 47.8% | 52.2% |

**Temporally-Adaptive RoPE.** We also explore different approaches of the RoPE design in temporal dimension for temporal alignment in ReactID. Two additional runs of attention masking and vanilla RoPE are conducted for comparison. **Attention masking** directly applies the temporal mask on video tokens to enforce the timeline prompt only attend to the corresponding frames, and there is no temporal RoPE implemented. **Vanilla RoPE** assigns original temporal RoPE to video tokens without temporal axis rescaling. In addition to utilize the metric of CLIP-T, we further leverage two measures from VideoScore (He et al., 2024a), i.e., Text-to-Video Alignment (TVA) and Temporal Consistency (TC) for temporal alignment evaluation. The performances of the three runs are summarized in Table 4. Compared to attention masking or vanilla RoPE that totally abandons RoPE or takes no account of event duration variances in temporal RoPE modeling, our temporally-adaptive RoPE dynamically adjusts the attention bias according to the sub-action duration in RoPE formulation, thereby achieving the best performances in temporal alignment. We also detail the performance comparisons by using different $T$. In practice, our ReactID yields the best results when $T$ is 2.

**Data Annotations.** In Table 6, We quantify the impact of annotations in ReactID-Data by evaluating four model-data combinations on ReactID-Eval-SEQ. **Base** refers to the ReactID model without timeline-specific modules, while **ReactID⁻** denotes the ReactID-Data without timeline annotations. Base+ReactID⁻ outperforms Base+OpenS2V-5M, confirming that our data pipeline drives generation quality. Notably, naively incorporating timeline information as enhanced prompt (Base+ReactID-Data) yields negligible gains, suggesting the **Base** struggles to interpret complex temporal contexts from text alone. In contrast, our full model (ReactID+ReactID-Data) achieves the optimal performance, indicating that our timeline-related design is crucial to fully leverage the timeline information for accurate sub-action generation.

**Subject Mask.** To verify our carefully-minted subject mask generation, we conduct a user study on three methods: LISA, vanilla SAM and our proposal. For each mask, participants select the best one. As shown in Table 7, ours achieves the highest preference rate. Details are in Appendix E.1.

**Timeline Annotation.** We compare the quality of timeline annotation generated by ReactID and other methods via user study. As indicated by Table 7, the F1 score attained by ReactID is high than those of other methods, verifying our better quality. Please refer to Appendix E.2 for more details.

**LLM-Planned Timelines.** To evaluate the quality of our LLM-planned timelines, we conduct two user study, i.e., user preference (Table 8) across three dimensions (Action Order, Logical Coherence and Motion Naturlness) and distinguishability (Table 9) between human expert and LLM. The quality of videos generated on LLM-planned timelines are comparable to that of using expert-writing timelines, verifying the reliability of ours. More details are in Appendix E.3.

## 5 CONCLUSIONS

We have presented ReactID, a comprehensive framework devised to address the fundamental tension between identity preservation and action realism in personalized video generation. We identified and tackled the root causes of this problem, i.e., imprecise data alignment, unstable training, and inadequate action modeling, through a holistic approach. Our solution integrates three key components: a high-quality dataset (ReactID-Data) with reliable identity-video correspondence, a progressive training curriculum for stable learning, and a novel timeline-based conditioning mechanism for fine-grained action control. Extensive experimental results validate that ReactID achieves a new state-of-the-art performances, excelling in both maintaining subject identity and generating natural, complex motions. Beyond its immediate performance gains, we believe this work makes two pivotal contributions to the field. First, the release of the ReactID-Data dataset provides a critical foundation for future research, offering a benchmark for precise identity-action modeling. Second, our timeline-based paradigm demonstrates a scalable and intuitive path toward controllable, multi-action video synthesis, significantly broadening the potential applications of personalized generation.

ACKNOWLEDGMENTS

This work was supported by the Key Science & Technology Project of Anhui Province No. 202523o09050002 and the National Natural Science Foundation of China under grants 62472399.

## 6 ETHICS STATEMENT

The primary of this paper is to introduce a personalized video diffusion model for reference subjects to perform desired action while maintaining the identity across frames. In this work, we construct our ReactID-Data from publicly available academic datasets. To ensure ethical compliance, we strictly adhere to licensing policies and respect privacy rights. We only utilize publicly accessible videos that are either explicitly licensed for research use or fall under fair-use considerations. No copyrighted content is redistributed or modified in violation of licensing terms. We uphold the highest ethical standards in the construction, use and dissemination of data.

## 7 REPRODUCIBILITY STATEMENT

The paper provides all necessary implementation details required to reproduce the main experimental results, including dataset processing procedure in Section 3.1 and Section B, model architectures, training protocols in Section 3.2, hyperparameters and experimental settings in Section 4.1. Additionaly, we will release our constructed dataset for academic research, ensuring faithful reproduction of our core results.

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

## A  APPENDIX: CLARIFICATION OF USING LLMS

In the period of paper writing, we only exploit the Large Language Model (LLM), i.e., GPT-4, for word polishing to improve the presentation quality. No LLMs contributes to the core scientific methods, technical contents, or any substantive research contributions.

## B  APPENDIX: DETAILS OF REACTID-DATA CONSTRUCTION

### B.1  VIDEO DATA PREPARATION

We first collect a pool of 20 million videos from publicly available video datasets such as HD-VG-130M (Wang et al., 2025) and OpenHumanVid (Li et al., 2025a). To detect the scene transition and partition long videos into single shot video clips, we utilize the PySceneDetect (Team, 2024b) to coarsely find the cut points according to color and brightness variations. Then, we detect the potential semantical transition points in videos using the cosine similarity of DINOv2 (Oquab et al., 2023) features between adjacent frames. Video clips longer than 5 seconds are saved to files using H.264 codec. To remove the text overlays, subtitles, and watermarks from videos, we use PaddleOCR (Team, 2024a) to find all text regions throughout the video, then crop it to the largest possible text-free area. Next, we apply a suite of automatic video quality assessment methods to evaluate each video clip across multiple facets, including aesthetic quality, technical quality, and motion dynamics, and filter out the clips with low scores in any of these dimensions. To provide rich, subject-centric textual captions which are crucial for personalized video generation, we employ an off-the-shelf vision-language model (Qwen2.5-VL (Bai et al., 2025)), which is prompted to analyze each subject appearing in the video and describe their physical appearance, corresponding actions, and background environment (Liu et al., 2025a). According to the descriptions, video clips that possibly contain an excessive number of subjects (e.g., a crowd of people or a line of cars) are filtered out, ensuring that the remaining videos clear focus on a limited number of subjects.

**Video Scene Cut.**  To prepare a high-quality video clip pool, we employ a systematic two-stage strategy to partition long videos into semantically coherent clips. The process begins with a coarse, content-aware segmentation using PySceneDetect. Subsequently, we refine these initial segments with DINOv2, inserting additional cuts at temporal locations where the cosine similarity between adjacent frame features drops below a predefined threshold. Finally, we retain all clips with durations above 5 seconds, which are well-suited for personalized video generation model training.

**Video Quality Filtering.**  We implement a multi-dimensional video quality scoring mechanism:

(1) **Aesthetic Score:** We utilize an improved aesthetic predictor to evaluate the visual appearance of each clip, only retaining those with high aesthetic scores.

(2) **Motion Score:** To distinguish deliberate subject motion from incidental camera movement, we compute optical flow using RAFT model. The foreground and background flows are calculated separately, and the final score is based on the foreground flow, effectively isolating and quantifying subject-specific motion.

(3) **Content Purity:** We employ PaddleOCR to detect and remove frames containing overlaid text or watermarks. Frames with OCR confidence scores above 0.7 are flagged, and the clip is cropped to retain only clean visual content.

(4) **Technical Quality:** We assess the technical quality using DOVER model for data filtering.

**Video Captioning.**  Effective captions for personalized video generation must be subject-centric, detailing the reference subject's appearance and motion while incorporating relevant contextual cues from the background environment. To meet this requirement, we employ Qwen2.5-VL-7B to generate rich descriptions. Moreover, we filter out any captions containing plural entity nouns. This operation not only maintains a clear focus on a limited number of subjects, but also facilitates consistency between the used textual prompts during training and inference.

### B.2 DATASET ANNOTATION

We further construct the timeline annotation based on the collected video data. There are four main steps in this data processing stage, i.e., entity label extraction, subject detection and segmentation, face detection and segmentation, and timeline prompt captioning.

**Entity Label Extraction.** To accurately identify distinct entities, we adopt a hybrid strategy that leverages both textual and visual information for entity labeling, mitigating the risk of error propagation from the video recaptioning. (1) For object label extraction, a text-based approach is exploited, which first pre-defines a taxonomy of common objects, and then use a NER model, i.e., SpaCy (Honnibal & Montani, 2017), to parse the captions and extract relevant entity labels in the taxonomy. (2) To deal with human and living subjects, we shift to a vision-based method to accurately discriminate entities with similar semantics. We leverage Qwen2.5-VL model to analyze the video content and extract unique, descriptive entity labels. The potential inaccuracies caused by the generated captions can be alleviated, providing a more robust foundation for subsequent subject and face detection.

**Subject Detection and Segmentation.** Conventional CLIP-based detectors like GroundingDINO (Liu et al., 2024) can fail to distinguish semantically similar concepts due to the limitation in understanding. To overcome this, we leverage an MLLM-based detector, i.e., Florence-2 (Xiao et al., 2024), for highly accurate subject detection. To further verify label-subject correspondence, we perform a subsequent classification step using SigLIP (Zhai et al., 2023) as post-verification (Zhang et al., 2023). Finally, we apply SAM2.1 (Ravi et al., 2024) to generate segmentation masks for the detected subjects. These masks are further refined using morphological operations to improve boundary precision.

**Face Detection and Segmentation.** To detect high-quality human faces, we first utilize Insightface toolbox on video frames with 2 FPS sampling rate. Then, we apply Non-Maximum Suppression with a 25% overlap threshold to filter duplicates and remove outlier bounding boxes. For each unique person, the detected faces are ranked by confidence, and five representative frames are uniformly sampled. Finally, SAM2.1 model is employed to generate precise masks for the frame with highest detection confidence, which are subsequently refined using morphological operations as well.

**Timeline Prompt Captioning.** We devise an ensemble approach that combines the vision language model (VLM), i.e., Qwen2.5-VL-72B (Bai et al., 2025), with two visual temporal grounding models UniMD (Zeng et al., 2024) and TFVTG (Zheng et al., 2024) to obtain both timestamps and the descriptions of sub-actions (Tan et al., 2026). First, we instruct Qwen2.5-VL to identify and caption each sub-action in the video based on meaningful action changes and detect the timestamps of each sub-action. Next, utilizing these captions of each sub-action as queries (Zeng et al., 2025), we employ UniMD and TFVTG to independently re-localize the temporal boundaries. This process yields three distinct sets of candidate annotations per sub-action: one originating directly from the Qwen2.5-VL and two derived from specialized visual temporal grounding models. Finally, we employ InterVideo2 (Wang et al., 2024a) as a scoring model. For each candidate, we utilize the video and text encoders of InterVideo2 to extract visual and textual representations from the video segment and the sub-action caption, respectively. We then calculate the cosine similarity between these embeddings as alignment score. The candidate timeline annotation (comprising both the timestamp and caption) with the highest alignment score is selected as the final detection for each sub-action.

## C APPENDIX: STATISTICAL ANALYSIS ON REACTID-DATA

Figure 7 further demonstrates a statistical analysis on our proposed ReactID-Data. We measure the video dataset from the perspectives of duration, resolution, word sequence length of prompt, aesthetic score, and motion score. The word cloud of entity label is also depicted to show the major components of the detected subjects. According to the statistics, the duration of more than 97% videos is longer than 5 seconds, and all videos have a resolution of at least 720P, guaranteeing the high video quality for model training. Meanwhile, around 94% videos having $\geq 50$ words in each text prompt indicate that ReactID-Data can provide precise and comprehensive textual descriptions for videos to enhance video generation learning. The distribution of motion score is very uniform, reflecting that the training data can cover both relative static and dynamic scenes. In the word cloud, the entities such as human (e.g., woman and man), animals (e.g., horse) and common objects (e.g., sofa and mug) have been included. The reasonable data composition of ReactID-Data highlights the merit for precise personalized video generation modeling.

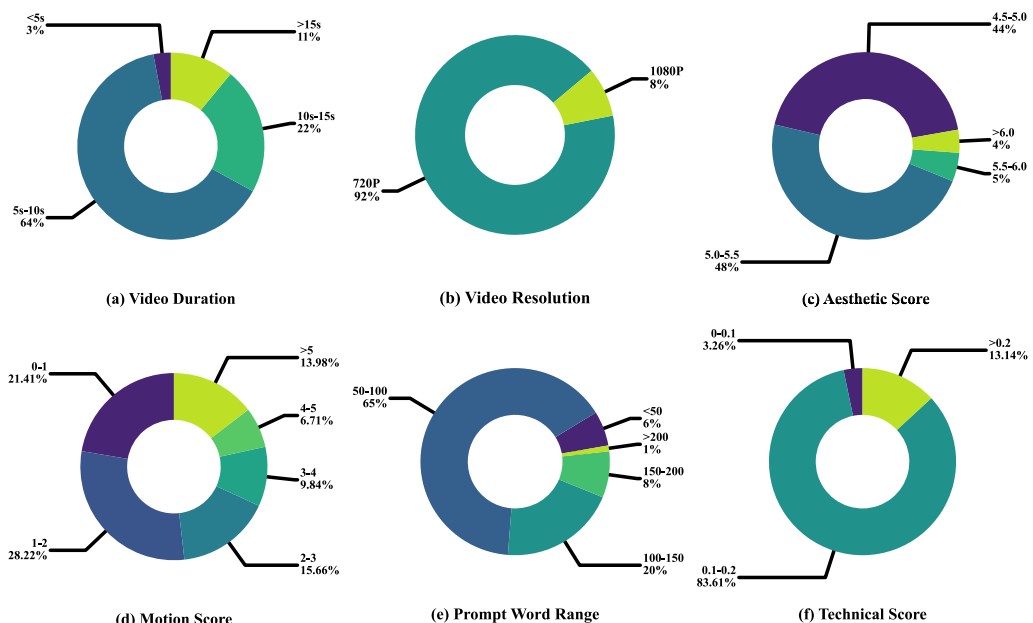

Figure 7: Statistical analysis on our ReactID-Data.

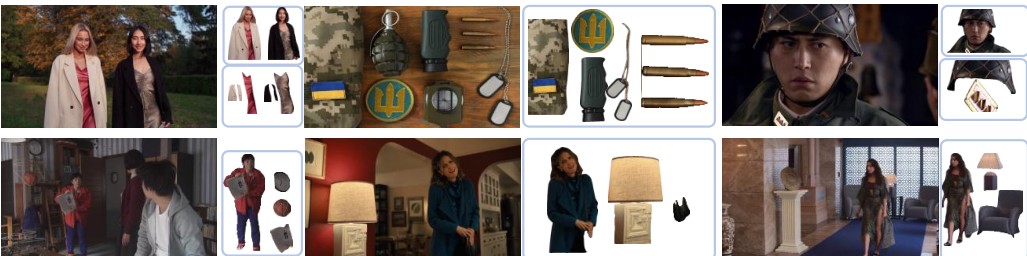

Figure 8: Examples of data items in the ReactID-Data.

# D  APPENDIX: OVERALL ARCHITECTURE OF REACTID

Given the subject-to-video pairs with timeline annotations, we proceed along the direction of typical personalized video generation DiT framework for subject condition injection, and further consider the timeline information learning. We choose the Wan2.1-1.3B (Wan et al., 2025) as the base architecture of our ReactID. To preserve the intrinsic capabilities of the original DiT blocks while enabling personalized video generation, we introduce a dedicated conditioning branch for reference image feature encoding. This conditioning branch consists of several reference blocks that are structurally identical to the base DiT blocks. We first apply the VAE on the reference images to obtain the image latents, which are treated as the condition and further fed into the condition branch. The extracted middle-level features from the image condition are then injected into the video diffusion model via Res-Tuning (Jiang et al., 2023) manner. To handle the subject timeline integration in diffusion, we introduce a novel subject-synchronized module which consists of two specific designs, i.e, subject-aware cross-attention and temporally-adaptive RoPE, in DiT blocks. The former ensures robust subject-action association in multi-subject scenarios by accurately binding each action to its corresponding subject. The latter precisely anchors each sub-action within the timeline to its designated timestamps, enhancing temporally alignment and semantically coherence.

# E   APPENDIX: MORE EVALUATION

## E.1   HUMAN EVALUATION ON SUBJECT MASK GENERATION

To obtain more accurate subject mask for our ReactID training, we improve the performance of vanilla SAM (Ravi et al., 2024) segmentation pipeline by leveraging additional visual foundation models and VLMs. First, we utilize SigLIP (Zhai et al., 2023) to filter out bounding boxes with low image-text similarity to ensure that SAM receives correctly matched subject region for segmentation. Then, we apply iterative morphological operations, including dilation and hole filling to improve mask continuity. Finally, we employ Qwen2.5-VL-32B to score and verify the final mask quality.

To validate the effectiveness of our deliberate design, we conduct a human evaluation on 200 randomly sampled cases. We recruit 20 participants from diverse backgrounds to perform a blind, side-by-side comparison across three segmentation approaches: LISA (an MLLM-based segmentation method), vanilla SAM and our method. For each sample, participants select the best mask. As shown in the right part of Table 7, our pipeline achieves the highest preference rate, demonstrating its effectiveness in reducing segmentation failures in challenging real-world scenarios.

## E.2   HUMAN EVALUATION ON TIMELINE ANNOTATION

We conduct a user study to compare the quality of timeline annotation between our ReactID and other VLMs or specialized models. Particularly, 20 participants are invited from diverse academic and professional backgrounds. We randomly sample 200 timeline annotations from our dataset for the human evaluation. Each participant is instructed to evaluate every annotation generated by ReactID and other VLMs along two dimensions, event and timestamp accuracy. According to the evaluation results, we classify the timeline annotations into three categories to compute the F1 score.

• False Negatives: When the annotation pipeline misses a salient event present in the video.

• False Positives: When the detected event does not exist in the video or the timestamp does not match the event boundaries.

• True Positives: When both the caption and timestamp are correct.

As shown in the left part of Table 7, our ReactID achieves a superior F1 score of $0.78$, supported by a inter-annotator agreement of $0.86$. The Qwen2.5-VL follows as the second-best annotator with an F1 score of $0.71$. The results reflect the robustness of our timeline generation pipeline, and the precision of the timeline annotations to facilitate timeline-based generation model training.

## E.3   HUMAN EVALUATION ON LLM-PLANNED TIMELINES

To evaluate the quality of our LLM-planned timelines for personalized video generation, we conduct two types of human evaluations, i.e., user preference and distinguishability, between the timelines generated by human expert and LLM. Specifically, we instruct 20 participants to evaluate 200 videos generated on the human expert-annotated and our LLM-planned timelines across three dimensions: Action Order (AO), Logical Coherence (LC), and Motion Naturalness (MN).

**User Preference.** The results in Table 8 demonstrate comparable human preference between the generated videos using such two kinds of timelines. Notably, videos conditioned on LLM-planned timelines even slightly outperform expert annotations in terms of Motion Naturalness, suggesting superior fluidity in the generated content.

**User Distinguishability.** Table 9 further presents the confusion matrix of the human distinguishability test. As observed, evaluators could not reliably differentiate between the human-annotated or our LLM-based sources. The evaluation indicate that our timeline annotation planned by LLM is reliable and aligns closely with the human expert logic.

## E.4   COMPARISON BETWEEN OPENS2V-5M AND REACTID-DATA

We conduct a comprehensive comparison between recent public OpenS2V-5M (Yuan et al., 2025a) and our proposed ReactID-Data. First, we randomly select $2,000$ samples from both OpenS2V-5M and ReactID-Data, then employ CLIP to classify subject images against their corresponding

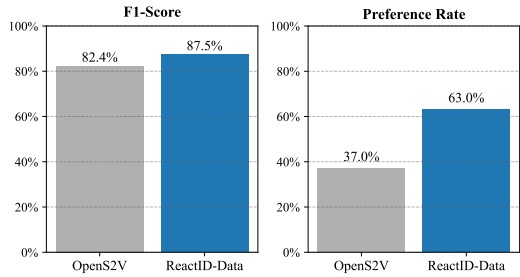

Table 10: Comparisons of ConsisID trained on different datasets. Models are evaluated on the ReactID-Eval-SEQ dataset.

| Training Data | FaceSim ↑ | Gme. ↑ | Natural. ↑ |
|---|---|---|---|
| OpenS2V-5M | 45.12% | 72.86% | 59.51% |
| **ReactID-Data** | **45.41%** | **73.20%** | **64.77%** |

Figure 9: Classification and preference comparison between OpenS2V-5M and our ReactID-Data.

Table 11: Quantitative comparison of different approaches on our ReactID-Eval-SEQ dataset. The best-performing result is highlighted in **bold**.

| Method | Aes.↑ | M. Smo.↑ | M. Amp.↑ | FaceSim↑ | Gme.↑ | Nexus.↑ | Natural.↑ | Total.↑ |
|---|---|---|---|---|---|---|---|---|
| *Without Prompt Pre-processing* | | | | | | | | |
| VACE-1.3B | 47.92% | 92.73% | 23.65% | 19.48% | 70.02% | 35.18% | 64.78% | 48.58% |
| Phantom-1.3B | 45.68% | 92.16% | 15.38% | 37.41% | 67.35% | **38.69%** | 62.30% | 51.40% |
| *With LLM Enhanced Prompts* | | | | | | | | |
| VACE-1.3B | 48.20% | 92.60% | 23.45% | 19.70% | 70.00% | 35.30% | 64.90% | 48.70% |
| Phantom-1.3B | 45.80% | 92.05% | 15.25% | 37.20% | 67.50% | 38.60% | 62.40% | 51.30% |
| *With LLM Planned Structured Timeline Prompts* | | | | | | | | |
| VACE-1.3B | 47.98% | 92.78% | 23.38% | 19.25% | 69.96% | 35.05% | 64.52% | 48.25% |
| Phantom-1.3B | 45.75% | 92.10% | 15.18% | 37.05% | 67.45% | 38.50% | 62.05% | 50.95% |
| **ReactID** | **49.11%** | **94.58%** | **39.46%** | **38.20%** | **71.23%** | 37.13% | **68.69%** | **54.42%** |

entity labels. We compute F1-score to evaluate classification performance. As shown in left part of Figure 9, ours achieves significantly higher F1-scores, demonstrating the superiority of our data construction pipeline. When handling complex scenarios containing entities with high visual or semantic similarity, our pipeline can maintain precise entity labeling and accurate subject extraction. However, the commonly adopted text-based entity extraction approach (i.e., Grounded-SAM) in OpenS2V may suffer from omissions and misclassifications. To further validate the labeling quality of the two measurements, we conduct a human evaluation. Specifically, we randomly sample 2,000 valid instances from the HD-VG-130M dataset and process them using both the conventional data pipeline and our data pipeline. 25 participants are recruited to evaluate the quality of both datasets, specifically assessing whether subject extraction is accurate, complete, and consistent with the assigned labels. Each participant selects the better-processed result from the two alternatives. We report the average preference rate across all trials in the right part of Figure 9. There are 63% of participants that favor our data processing pipeline, compared to only 37% for the conventional approach. To further testify the quality of ReactID-Data, we train ConsisID from scratch under identical setting on 500K samples randomly selected from the OpenS2V-5M and ReactID-Data, respectively. As shown in Table 10, the model trained on ReactID-Data outperforms the model trained on equivalent data in OpenS2V-5M, confirming that our ReactID-Data facilitates superior model optimization in personalized video generation.

### E.5 MODEL PERFORMANCES ON DIFFERENT PROMPT FORMATS

To analyze whether the existing models can be improved through enriched text prompts, we select two top-performing models, VACE-1.3B and Phantom-1.3B, and perform these two methods on the input prompts processed by two augmentation strategies: prompt enhancement and structured timeline prompts. The performances on OpenS2V are illustrated in Table 11. Prompt enhancement yields minimal performance gains across all baselines that generate videos without any prompt preprocessing. The results are expected since the majority of prompts in the OpenS2V already consist of detailed, comprehensive captions. Notably, for the models trained without timeline annotations, the structured timeline prompt also fails to deliver significant improvements and, in some cases, leads to performance degradation. This somewhat reveals a fundamental limitation: models not explicitly trained on timeline-structured data possess limited temporal reasoning capabilities, and cannot effectively leverage sequential action descriptions to enhance motion naturalness. Without designing

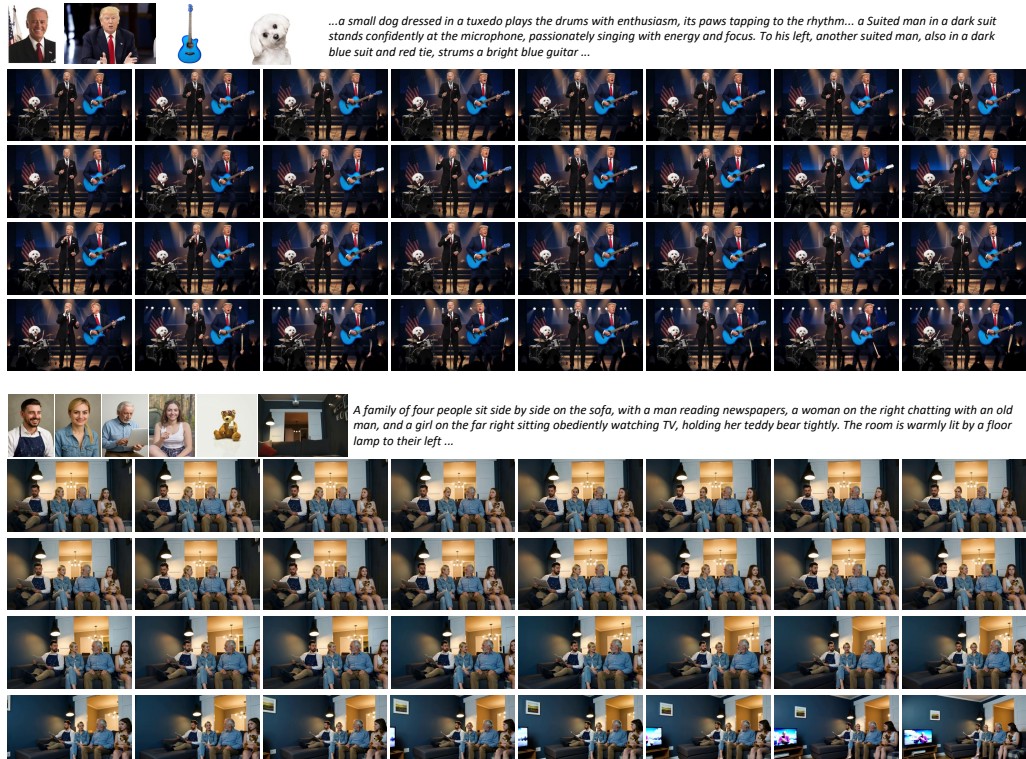

Figure 10: Visualization of two 30-second long videos generated by ReactID.

proper temporal modeling architectures, simply reformatting single text prompts into timeline structures may introduce confusion for action modeling in video generation. These findings underscore the importance of architectural design and training paradigms tailored for temporal action modeling, rather than solely relying on input formatting modifications.

### E.6 Long Video Generation

Long video generation in personalized video generation generally suffers from identity drift (Chen et al., 2025a; Luo et al., 2024), minimum or collapsed motion. Although ReactID was not explicitly optimized for long video generation, it can be effectively coupled with an auto-regressive generation strategy to generate longer videos while maintaining high identity consistency. Specifically, ReactID generates the video segment-by-segment (e.g., in 5-second clips), utilizing the preceding segments as the visual condition for the subsequent one. This iterative process allows us to customize videos extending well beyond the standard 5-second duration. As shown in Figure 10, we successfully generate long videos in multi-subject scenarios with four and six reference subjects while maintaining the identity of all subjects throughout the video, without significant temporal drift.

### E.7 Additional Visualization

We provide additional visualization to further demonstrate the capabilities of ReactID. As shown in Figure 11, given predefined timelines in which each subject performs distinct action sequences, ReactID faithfully follows the temporal schedule to generate a series of overlapping or asynchronous actions performed by different subjects. The results highlight the model's ability to handle complex temporal dynamics. As shown in Figure 12, when provided with only a single face reference image, ReactID preserves fine-grained identity cues while remaining well aligned with the text prompt. As illustrated in Figure 13, ReactID demonstrates strong robustness and generalization ability in multi-subject scenarios. It supports both human and non-human entities, including animals and common objects, and enables flexible combinations across different categories. Compared with other models, ReactID effectively mitigates the copy-paste issue and produces consistent subjects with coherent

and natural actions, striking a balance between identity preservation and action realism. Additional results in Figure 14 further validate the versatility and stability of ReactID across diverse scenarios involving single subject and multiple subjects. Our model is capable of generating Overall, our model is capable of generating high-fidelity subjects and realistic, natural interactions.

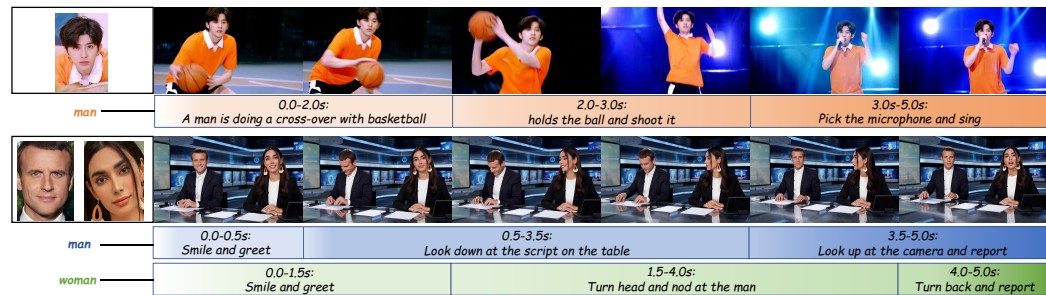

Figure 11: Additional results of ReactID conditioned on given timelines.

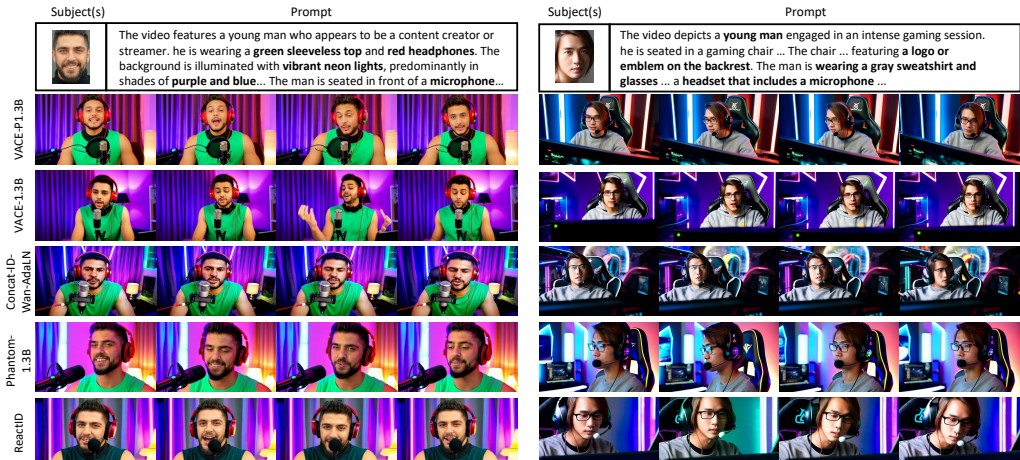

Figure 12: Additional comparative results of single-face customization across different methods.

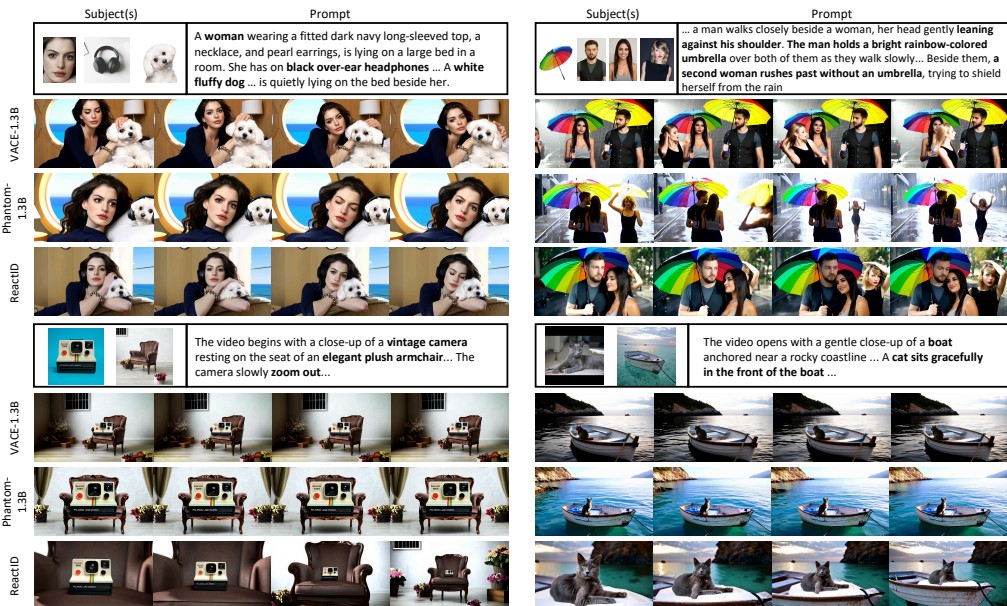

Figure 13: Additional comparative results of multi-subject customization across different methods.

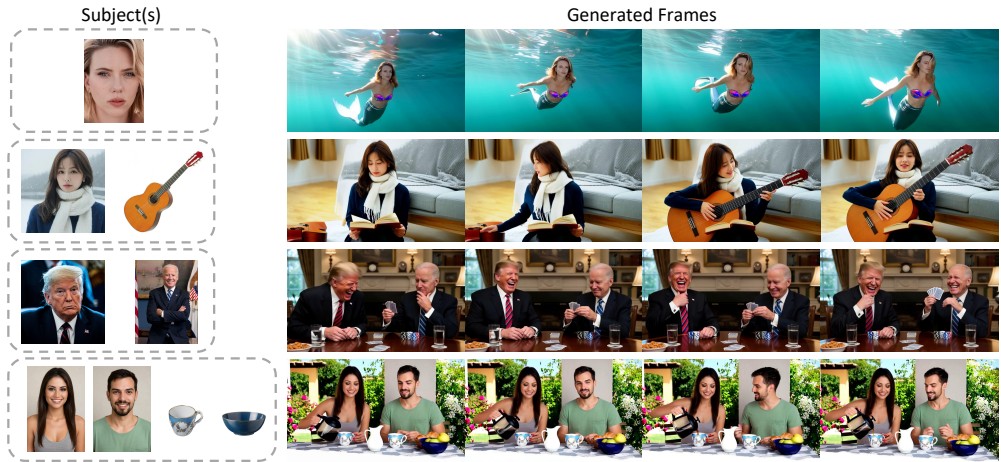

Figure 14: Additional results generated by ReactID.

