# OpenReview forum: "ReactID: Synchronizing Realistic Actions and Identity in Personalized Video Generation"
_ICLR.cc/2026/Conference — ICLR 2026 Poster_

### Official Review · Reviewer_GCZR · 2025-10-15

**Soundness:** 3
**Presentation:** 4
**Contribution:** 3
**Rating:** 6
**Confidence:** 4

**Summary:**

This paper presents ReactID, a comprehensive framework designed to address the fundamental trade-off between identity consistency and action realism in personalized video generation. The authors identify three core challenges—imprecise subject-video alignment, unstable training dynamics, and inadequate modeling of fine-grained actions—and propose a holistic solution. The main contributions are threefold: (1) the creation of ReactID-Data, a large-scale, high-precision dataset with detailed temporal annotations; (2) a novel Difficulty-Aware Curriculum Learning strategy to mitigate the "copy-paste" artifact and ensure stable convergence; and (3) a structured, timeline-based conditioning mechanism with two novel components (Subject-Aware Cross-Attention and Temporally-Adaptive RoPE) for fine-grained action control. The paper demonstrates through extensive experiments that ReactID achieves state-of-the-art performance, effectively balancing identity preservation with natural motion generation. It is a solid piece of work with significant practical contributions to the field.

**Strengths:**

1.  **Clear Motivation and Problem Formulation:** The paper begins with a clear and insightful analysis of the core challenges in personalized video generation. It methodically breaks down the problem into three specific bottlenecks. The proposed methods directly and logically address each of these points, creating a coherent and well-motivated narrative.

2.  **High-Quality Data Pipeline and Dataset:** The construction of the ReactID-Data dataset is a major strength. The data processing pipeline is thorough, robust, and well-reasoned, combining vision-based entity extraction, MLLM-based detection, and post-verification. The decision to annotate videos with structured action timelines and fine-grained textual descriptions is particularly valuable, addressing a critical need for temporally precise data in the field.

3.  **Effective Curriculum Learning Strategy:** The proposed Difficulty-Aware Curriculum Learning is an elegant and well-thought-out solution to the common "copy-paste" problem, where models overfit to easy samples at the expense of motion realism. By formalizing sample difficulty along three intuitive axes (subject size, appearance similarity, and sampling strategy) and progressively introducing harder samples, the method effectively guides the model toward a more generalized solution. This claim is convincingly supported by a thorough ablation study (Table 5), which demonstrates the value of each component and the full curriculum.

4.  **Novel and Controllable Timeline Conditioning:** The timeline-based conditioning mechanism is a significant step forward for controllable video generation. Decomposing monolithic prompts into timestamped sub-actions gives users granular control over complex action sequences. The architectural innovations—Subject-Aware Cross-Attention to bind actions to specific subjects and Temporally-Adaptive RoPE to handle variable action durations—are technically sound and well-justified.

5.  **State-of-the-Art Performance:** The experimental results are comprehensive, demonstrating superior performance against numerous baselines across multiple benchmarks (OpenS2V-Eval and the newly proposed ReactID-Eval-SEQ). The qualitative results (Figure 1, Figure 4) further underscore the model's ability to generate fluid, natural movements while faithfully preserving the subject's identity, a feat many prior methods struggle with.

**Weaknesses:**

1.  **Lack of Ablation on Data Annotation Quality:** While the paper shows that a model trained on ReactID-Data outperforms one trained on OpenS2V-5M (Table 6), it lacks a direct ablation study on the impact of the fine-grained timeline and textual annotations. It would be beneficial to see how the ReactID model itself performs when trained on the same video data but with coarser captions or no timeline promtps, which would more directly quantify the benefit of the expensive and detailed annotation pipeline.

**Questions:**

None

---

> ### Author Response · Authors · 2025-11-23
> **Response to Reviewer GCZR**
>
> Thank you for the constructive comments. We are encouraged that you think our method is innovative, insightful and well-motivated.
>
> **[W1] Ablation on Data Annotation Quality.**
>
> Thanks for the valuable suggestion. In view of this comment, we have added a detailed ablation study in **Table G** here (Table 6 of the revised manuscript) to reveal the direct effect of the high-precision video data and the timeline annotation separately. In summary, solely using the high-precision video data increases the TotalScore from 50.37% to 52.14%. Simply using timeline annotation as a text prompt barely brings any performance improvement, while our ReactID framework achieves a 2.28% increase in TotalScore through the injection of timeline annotation.
>
> **Table G: Ablation study of designs in ReactID w.r.t. timeline on ReactID-Eval-SEQ.**
> | Model       | Data                     | TotalScore |
> |-------------|--------------------------|:------------:|
> | Baseline    | OpenS2V-5M               | 50.37%     |
> | Baseline    | ReactID-Data w/o Timeline| 52.14%     |
> | Baseline    | ReactID-Data w Timeline  | 52.21%     |
> | ReactID     | ReactID-Data w Timeline  | 54.42%     |
>
> To quantify the benefit of our annotation pipeline, we compared the Baseline model (ReactID model with timeline-specific modules removed) trained on ReactID-Data w/o Timeline and OpenS2V-5M. As shown in the first two rows of **Table G**, the model trained on ReactID-Data w/o Timeline (52.14%) outperforms the one trained on OpenS2V-5M (50.37%). It confirms that our meticulously designed data pipeline is a primary driver of generation quality, even without explicit timeline modeling.
> To validate the use of timeline data, we experimented with naively incorporating timeline information as enhanced text prompts input. The model yields negligible gains (52.21%) over the baseline model trained without timeline (52.14%), which suggests that the baseline model struggles to effectively interpret complex temporal context solely from text. In contrast, our full ReactID model with timeline-related design trained on full ReactID data with timeline achieves the optimal performance (54.42%). This demonstrates that high-quality timeline annotations alone are not enough, our specific timeline-related design is crucial to fully unlock the potential of the fine-grained data.
> In conclusion, the ablation confirms that both the quality of our data annotations and the specialized model architecture are essential and complementary contributions to the final performance.
>
> Thank you again for your valuable suggestion. We believe these enhancements improve the presentation and strengthen the benefit of our data pipeline and model design.

---

### Official Review · Reviewer_3tVk · 2025-10-29

**Soundness:** 2
**Presentation:** 2
**Contribution:** 2
**Rating:** 4
**Confidence:** 1

**Summary:**

This paper addresses a challenge in personalized video generation: balancing identity consistency (preserving a subject’s unique features across frames) and action realism (generating natural, dynamic motions). The authors identify three root causes of this trade-off—imprecise subject-video alignment, unstable training due to varying sample difficulty, and coarse-grained action modeling—and propose ReactID, a holistic framework to mitigate them.

**Strengths:**

1. The problem of identity-action balance is central to personalized video generation, and ReactID’s solutions (dataset, curriculum, temporal modeling) advance the field beyond incremental improvements. The dataset release also fosters collaboration.
2. ReactID’s holistic approach (data + training + modeling) is innovative. For example, the subject-aware cross-attention’s label binding mechanism (assigning unique labels to subjects) solves a longstanding issue of action-subject misalignment in multi-subject scenarios.
3. Ablation studies validate each component.

**Weaknesses:**

1. The paper does not report training/inference time compared to baselines. For real-world use (e.g., edge devices), efficiency is critical—readers cannot assess if ReactID’s performance gains come at the cost of slower runtime.
2. ReactID uses LLMs to convert single prompts to timelines, but does not evaluate how LLM errors (e.g., incorrect action order, wrong timestamps) affect generation quality. This is a practical limitation, as LLMs are not perfect.

**Questions:**

1. How does ReactID perform on scenarios with 3+ subjects (e.g., a family of 4 performing distinct actions)? Does the label binding mechanism in subject-aware cross-attention scale efficiently, or does it introduce misalignment/overhead with more subjects?
2. Can you provide training time per step (on a standard GPU, e.g., A100) and inference time per 5-second video, compared to baselines ? This would help readers assess ReactID’s practicality.
3. ReactID focuses on 5-second videos—can it generate longer videos (e.g., 30 seconds) while maintaining identity-action consistency? If not, what are the main barriers (e.g., temporal drift)?

---

> ### Author Response · Authors · 2025-11-23
> **Response to Reviewer 3tVk (1/2)**
>
> Thank you for the constructive comments. We are encouraged that you think our data release fosters collaboration and our holistic approach is innovative.
>
> **[W1, Q2] Training/Inference time.**
>
> Thanks for your valuable suggestion. We have compared our runtime with the closest baseline VACE on NVIDIA A100 GPU(s), as summarized in **Table D** here.
>
> **Table D: Comparison of training and inference time between VACE-1.3B and ReactID.**
> | Method      | Training time per step (8 GPUs) | Inference time per video (1 GPU) |
> |-------------|:----------:|:-----------:|
> | VACE-1.3B   |   ~62.5s    | ~307s     |
> | ReactID     | ~65.8s (+5.3%)    | ~316s (+2.9%)     |
>
> Compared to VACE, ReactID incurs only a marginal overhead, with an increase of approximately 5.3% in training time per step and only 2.9% in inference time for a 5-second video generation. Overall, ReactID delivers performance improvements while maintaining practical runtime efficiency, with only minor additional computational cost. We have updated the Implementation Details in the revised manuscript to report the time budget.
>
> **[W2] Analysis of LLM errors in timeline conversion.**
>
> Thanks for pointing this out. Yes, the output of LLM is not perfect. In view of this comment, we have conducted a series of user studies to reveal if the possible LLM error affects the generation quality of our ReactID.
> The first user study is **human preference comparison between LLM-planned and human expert-planned videos**. We random selected 50 video captions and paired reference subject image from our dataset, and asked both GPT-4o and human experts to expand the captions into structured timelines. We instructed 5 human experts to manually craft 10 timelines per expert, resulting in 50 Expert-annotated timelines. Also, we prompted GPT-4o to generate structured timelines for the same 50 captions, resulting in 50 LLM-planned timelines. Using our ReactID model, we generated 2 videos for each timeline prompt along with reference images with different random seeds. This yielded a total of $200$ videos ($100$ from Expert timelines, $100$ from LLM timelines). We instructed participants to blindly evaluate these videos across three dimensions: Action Order (AO), Logical Coherence (LC), and Motion Naturalness (MN). As shown in **Table E**, the preference rates between two sets of generated videos are highly comparable, indicating negligible perceptual difference. Notably, videos generated from LLM-planned timelines achieved a slightly higher preference rate in Motion Naturalness, suggesting superior fluidity in the generated content.
>
> **Table E: Human preference comparison between videos generated from Expert and LLM-planned timelines across AO, LC, and MN dimensions.**
> | Dimensions | LLM | Expert |
> |--------|:-----:|:--------:|
> | Action Order       | 46% | 54%    |
> | Logic Coherence    | 48% | 52%    |
> | Motion Naturalness | 53% | 47%    |
>
> The second user study is **distinguishability of the source of timeline (from LLM or from human expert)**. Evaluators were asked to classify whether a given timeline annotation was produced by an LLM or a human expert. **Table F** presents the confusion matrix of this classification. The accuracy for both classes hovers near 50% (random guess), which confirms the timeline annotations generated by the LLM are virtually indistinguishable from those meticulously annotated by experts. While LLMs are not perfect, these two experiments demonstrate that the LLM-planned timelines are robust enough that they do not pose a bottleneck for generation quality.
>
> **Table F: Human Distinguishability of Timelines from Expert vs. LLM.**
> | GroundTruth \ Prediction | LLM   | Expert |
> |---------|:-------:|:--------:|
> | LLM     | 52.8% | 47.2%  |
> | Expert  | 47.8% | 52.2%  |
>
> These findings indicate that although LLMs may occasionally produce imperfect action orders or timestamps, such deviations are generally minor and do not materially affect the generation quality. Moreover, our global prompt guidance offers a degree of built-in tolerance to timeline noise. Taken together, the human study demonstrates that using an LLM for prompt-to-timeline conversion during inference is both practical and sufficiently reliable.

---

> ### Author Response · Authors · 2025-11-23
> **Response to Reviewer 3tVk (2/2)**
>
> **[Q1] Performance on 3+ subjects scenarios.**
>
> ReactID can generalizes robustly to scenarios with **three or more subjects**, including cases where each subject performs distinct and asynchronous actions. We have incorporated a challenging case in the right of **Figure 6** in the revised manuscript. The figure depicts a scenario involving **a family of 4 human subjects** (*Man in apron*, *Woman*, *Old Man*, *Girl*) and 1 object (*Teddy Bear*) and a scene (*Living Room*). The model successfully executes highly distinct, independent actions for all 4 subjects concurrently.
> The *Man* reads a newspaper ($0–3.5s$) then turn his head and talks ($3.5–5s$).
> The *Girl* looks right and holds a teddy bear ($0–3.5s$) then gazes ahead ($3.5–5s$).
> Simultaneously, the *Woman* and *Old Man* engage in a conversational action sequence ("Look right" followed by "Talk")
> The generated identities remain distinct and faithful to their respective reference images. There is no visual evidence of identity blending (e.g., the apron texture does not bleed onto the other subjects).
> The results validate that our label binding mechanism does not suffer from significant misalignment or degradation as the number of subjects increases (also see **Figure 5**). It scales efficiently since the binding process involves only constant label assignment operations, it does not incur additional computational overhead as the number of input subjects increases.
> Although ReactID readily supports generation with 3+ subjects, we have observed that such scenarios account for only a minimal proportion of our training data. The majority of video involve only a single subject or a small number of subjects. While our current model handles 3+ subject scenarios through its robust generalization capability, achieving better performance in this challenging setting in the future would require targeted augmentation of 3+ subjects video data. This remains a direction for our future exploration.
>
> **[Q3]: Longer video generation.**
>
> Yes. Although ReactID was not explicitly optimized for long video generation, it can be effectively coupled with an auto-regressive generation strategy to generate longer videos while maintaining high identity consistency. Specifically, ReactID generates the video segment-by-segment (e.g., in 5-second clips), utilizing the preceding segments as the visual condition for the subsequent one. This iterative process allows us to customize videos extending well beyond the standard 5-second duration. We have incorporated two 30-second video demonstrations in **Appendix Figure 11** of the revised manuscript to illustrate this capability. As shown in the figure, we successfully generate long videos in multi-subject scenarios with four and six reference subjects while maintaining the identity of all subjects throughout the video, without significant temporal drift. To further enhance ReactID's performance in long video generation, we plan to collect high-quality video data featuring long continuous shots and utilize them as supervision signals for future model training.

---

### Official Review · Reviewer_vCvz · 2025-10-30

**Soundness:** 2
**Presentation:** 3
**Contribution:** 2
**Rating:** 4
**Confidence:** 4

**Summary:**

This paper targets personalized video generation and argues that current methods face a persistent tension between identity fidelity and realistic action. The authors propose ReactID, a framework that couples a new data pipeline, a difficulty-aware curriculum, and a timeline-based conditioning mechanism. ReactID-Data is built with entity extraction for living and non-living subjects, MLLM-assisted subject detection and verification, and subject masks for faces, aiming to deliver precise subject–video correspondences. The training strategy scores sample difficulty using subject size, appearance similarity, and sampling strategy, and then increases a difficulty threshold over time. The model introduces a subject-aware cross-attention with label binding and a temporally-adaptive RoPE that rescales time within sub-actions so temporal bias aligns with action boundaries. Experiments on OpenS2V-Eval and a new ReactID-Eval-SEQ show consistent gains on aggregated metrics and ablations indicate each component contributes. The paper also describes an LLM planner that converts a single prompt into a timeline at inference.

**Strengths:**

1. The paper is well scoped around balancing identity fidelity and action realism, and it articulates the root causes spanning data noise, training instability, and coarse action modeling.

2. The data pipeline is thoughtfully designed, separating living and non-living entities, grounding with MLLMs, and adding face-centric masks, which is important for identity preservation.

3. The timeline-based conditioning is technically interesting: subject-aware cross-attention with label binding plus temporally-adaptive RoPE to normalize sub-action durations targets a known failure mode at action boundaries.

**Weaknesses:**

1. The timeline annotations are produced automatically by a VLM at scale, yet the paper does not quantify timestamp accuracy or action-boundary noise, which could systematically bias training and evaluation for sequence control. A small human audit or agreement study would help.

2. Subject masks are supervised by SAM-style masks. Failure modes such as strong occlusion, rapid camera motion, or multi-subject overlap are not analyzed, although these are common in personalized scenarios.

3. Curriculum details are under-specified. The schedule for the difficulty threshold, the mixing ratio of intra-clip and inter-clip references over time, and the sensitivity to the λ weights are not provided, limiting reproducibility beyond the high-level recipe.

**Questions:**

1. How accurate are the VLM-derived timelines, both in absolute timestamps and boundary placement between sub-actions. Please report a small human study with boundary F1 and inter-annotator agreement.

2. Please clarify the compute budget, number of GPUs, and wall-clock time for the reported 10k-step training, and whether the method remains effective when scaled up or down.

---

> ### Author Response · Authors · 2025-11-23
> **Response to Reviewer vCvz (1/2)**
>
> Thank you for the constructive comments. We are encouraged that you think our data pipeline is thoughtfully designed and our timeline-based conditioning is interesting.
>
> **[W1, Q1] Evaluation on timeline annotations.**
>
> Thanks for the valuable suggestion. Yes, the quality of timeline annotations is important for our model training. Actually, to ensure the annotation quality, we have implemented a powerful multi-step multi-model annotation pipeline.
> This pipeline exploits two vision language models (Qwen2.5-VL-72B and InterVideo2) with two visual temporal grounding models (UniMD and TFVTG). Specifically, Qwen2.5-VL-72B first identifies sub-actions within the video, generating corresponding captions and timestamps for each. Subsequently, the two grounding models utilize these captions to independently generate two additional timestamp candidates per action. Finally, we employ InterVideo2 as a scoring model to evaluate the three candidate timestamps from distinct sources, selecting the one with the highest alignment score as the final annotation (comprising both the timestamp and caption).
> To compare our annotation pipeline with other single model baseline, here we conducted a rigorous human study involving $20$ participants from diverse backgrounds. We randomly sampled $200$ video clips from ReactID-Data and obtain four distinct timeline annotations for each clip: one from ReactID-Data, and three additional baselines generated individually by TFVTG, InternVL3-78B, and Qwen2.5-VL-72B. Participants were instructed to evaluate each timeline annotation based on event precision and timestamp accuracy to calculate the F1 score. As shown in **Table A** here, the ReactID annotations achieved the highest F1 score of $0.78$, supported by an inter-annotator agreement of $0.86$, significantly outperforming the standalone Qwen2.5-VL-72B baseline (F1 score of $0.71$). These results indicate the high quality of our timeline data. Although the timeline annotations obtained this way are still not perfect. However, our annotation quality is already top-tier in the industry, and as evidenced by the subsequent training results, it is robust enough to effectively support the training of our model.
>
> **Table A: Quality of the timeline annotations in ReactID-Data.**
> | Method | TFVTG | InternVL3-78B | Qwen2.5-VL-72B| ReactID|
> |:-:|:-:|:-:|:-:|:-:|
> |F1 Score| 0.57 | 0.63 | 0.71 | 0.78|
>
> We have updated the detailed annotation process and evaluation results in Section 3.3.1, Section 4.3, and Appendix B.2 of the revision.
>
> **[W2] Robustness of Mask Annotation.**
>
> Thanks for the valuable suggestion. Yes, the subject mask from SAM is not perfect. Considering this, when constructing our data, we have exploited a **multi-stage filtering and refinement process**. Specifically, for each detected subject, we first utilize SigLIP to calculate the similarity between the bounding box region and its corresponding text label; only regions with high semantic similarity are passed to SAM for segmentation. Subsequently, we apply a series of morphological transformations, including dilation and hole filling, to enhance mask continuity. Furthermore, we employ Qwen2.5-VL-32B to verify whether the generated mask correctly identifies the subject pixels. Through this process, samples with potential mask quality issues are filtered out to minimize training risks.
> We have compared our mask annotation process with baselines LISA and Vanilla SAM through human study. We randomly selected $200$ video clips and extracted one frame containing the subject from each to generate masks using the three methods. $20$ participants were then asked to select the mask that most completely and precisely identified the subject region for each case. As presented in **Table B** here, the data shows that our method achieved the highest human preference rate, suggesting that the pipeline effectively handles scenarios where standard models might be less robust.
>
> **Table B: Quality of the mask annotations in ReactID-Data.**
> | Method | LISA | Vanilla SAM | ReactID|
> |:-:|:-:|:-:|:-:|
> |Preference Rate| 15% | 31% | 54% |
>
> Detailed descriptions of the mask annotation process and the experimental results have been incorporated into Section 4.3 and Appendix E.1 of the revised manuscript.

---

> ### Author Response · Authors · 2025-11-23
> **Response to Reviewer vCvz (2/2)**
>
> **[W3] Specification of curriculum learning details.**
>
> Thanks for pointing this out. We have clarified the specific schedules and parameters below:
> We initialize the difficulty threshold $\tau$ at $0.53$. As training progresses, we update $\tau$ to $0.67$, $1.44$, and $1.84$ at training steps 5k, 7.5k, and 8.75k, respectively. After each update, the threshold remains fixed until the next scheduled change. The thresholds are derived from the statistical quartiles (25%, 50%, 75%, 100%) of the difficulty score ($D_{overall}$) distribution in our pre-sorted training set. This ensures the model is progressively exposed to harder samples in a statistically balanced manner. The mixing ratio of intra-clip and inter-clip samples is dynamically modulated by the difficulty threshold $\tau$. The curriculum learning naturally starts with a dominance of intra-clip samples and progressively introduces more challenging inter-clip samples as $\tau$ increases. Specifically, corresponding to the three update points of the threshold $\tau$, the proportion of inter-clip references within the total training samples transitions from an initial 0% to 0%, 11%, and 33%, respectively. Regarding $\lambda$ weights, we set $\lambda_{sub}=0.5$, $\lambda_{app}=1$, and $\lambda_{sam}=1$ according to the relative importance of each dimension. We observed that the model performance remains stable across a reasonable range of these values, meaning our proposal is insensitive to $\lambda$ weights.
> We have revised Section 3.2 in the manuscript to include these specifications to ensure reproducibility.
>
> **[Q2_1] Clarification of compute budget.**
>
> We conducted the 10k-step training on 8 NVIDIA A100 GPUs for around 183 hours, approximately 1464 GPU hours.
>
> **[Q2_2] Scalability.**
>
> Thanks for the valuable view point. We initially implemented ReactID based on Wan2.1-T2V-1.3B, which is a compact yet well-performing model that has been widely adopted by researchers and validated the superiority of our method based on it. To evaluate the potential of ReactID when scaled up, we have trained a preliminary version based on Wan2.1-T2V-14B for 3k steps. The results compared with model trained on Wan2.1-T2V-1.3B are summarized in **Table C**:
>
> **Table C: Comparison of TotalScore evaluated on the ReactID-Eval-SEQ between ReactID-1.3B and ReactID-14B across different training steps.**
> |     Model     |  1500  |  2000  |  2500  |  3000  |  4000  |  6000  |  8000  | 10000 |
> |-------------|:------:|:------:|:------:|:------:|:------:|:------:|:------:|:-----:|
> | ReactID-1.3B  | 29.11% | 30.87% | 32.56% | 36.67% | 42.25% | 49.17% | 53.08% | 54.42% |
> | ReactID-14B   | 31.06% | 33.56% | 35.77% | 40.81% |   -    |   -    |   -    |   -    |
>
> At equivalent steps, the 14B model significantly outperforms the 1.3B model in total score, demonstrating the scalability of our method. Notably, the 1.3B model continues to improve substantially beyond 3k steps. We can expect the 14B model would continue to show considerable improvements with further training. We will continue to train ReactID-14B and report the final performance as a scaled-up version of ReactID.

---

### Official Review · Reviewer_PVcC · 2025-11-01

**Soundness:** 3
**Presentation:** 3
**Contribution:** 3
**Rating:** 6
**Confidence:** 5

**Summary:**

This paper tackles the central trade-off in personalized video generation between identity consistency and action realism. The authors propose ReactID, a framework that coordinates advances in data curation, training strategy, and action conditioning to balance identity fidelity and motion naturalness. They introduce ReactID-Data, a large-scale dataset built via a high-precision pipeline. To stabilize learning, they analyze difficulty along axes such as subject size, appearance similarity, and sampling, and adopt a progressive curriculum from easy to hard to mitigate identity overfitting and copy-paste artifacts. For action modeling, they propose a timeline-based conditioning scheme that augments text prompts with structured multi-action sequences annotated with timestamps, integrated through two components: a subject-aware cross-attention module and a temporally-adaptive RoPE mechanism. Experiments report state-of-the-art performance on both identity preservation and action realism, suggesting the method effectively balances these competing objectives.

**Strengths:**

1. The manuscript is clearly written and easy to follow.
2. The motivation is well defined, pinpointing three key challenges in video customization: inaccurate identity preservation, unstable convergence, and compromised action naturalness.
3. The work is solid, introducing a new dataset and method components, including Subject-Aware Cross-Attention and Temporally-Adaptive RoPE.
4. Extensive experiments convincingly demonstrate the effectiveness of the approach.

**Weaknesses:**

1. Will the dataset, code, and model weights be released? Open-sourcing would significantly benefit community research and reproducibility.
2. How well does the method generalize to multi-subject customization beyond two subjects (e.g., three or more)?
3. For multi-subject scenarios, can the approach handle more complex, independent action timelines per subject (e.g., multiple subjects performing distinct actions concurrently)?
4. The results shown in the paper are about human subjects. Can the method support customizing animals and general objects?
5. Please consider citing closely related work:
    - CustomCrafter: Customized Video Generation with Preserving Motion and Concept Composition Abilities
    - ReVersion: Diffusion-Based Relation Inversion from Images
    - DreamRelation: Relation-Centric Video Customization

**Questions:**

See Weaknesses

---

> ### Author Response · Authors · 2025-11-23
> **Response to Reviewer PVcC (1/2)**
>
> Thank you for the constructive comments. We are encouraged that you think the proposed method is clearly motivated and well-presented.
>
> **[W1] Open-source of ReactID.**
>
> Yes, we will release the data, pre-trained model and code of ReactID, and we have already initiated the internal preparation for the open-source:
> - **ReactID-Data:** We will release the comprehensive dataset assets, including video clips, subject labels, bounding boxes, segmentation masks, and timeline annotations, alongside detailed documentation and preprocessing scripts. This ensures that community researchers can efficiently access ReactID-Data and make it readily available for model training.
> - **Model & Code:** We will release the pre-trained ReactID model weights together with the codebase. This will allow researchers to readily reproduce our results and utilize ReactID as a baseline for future studies.
>
> **[W2] Generalization to multi-subject customization.**
>
> Yes, ReactID supports customization for scenarios involving more than two subjects, and we demonstrate this capability in **Figure 5 (right)** and **Figure 6** of the revised manuscript.
> This capability is driven by our fine-grained annotation pipeline, which assigns discriminative labels to distinct entities within a clip, providing the rich supervision necessary for multi-subject customization.
> **Figure 5 (right)** and **Figure 6** of the revised manuscript showcase generation results for diverse settings including "3-humans", "2-humans + 1-object", and "4-humans + 2-objects + scene". As evidenced by these figures, ReactID maintains high identity consistency for three or more subjects while faithfully adhering to the provided text prompts and action timelines.
>
> **[W3] Complex and independent action timelines.**
>
> Yes, ReactID effectively handles video generation conditioned on multiple subjects, where each subject performs distinct actions concurrently. We demonstrate this in **Figure 6** of the revised manuscript.
> This capability is grounded in our data and model design: our annotation scheme assigns precise timelines to individual subjects, while the Subject-Aware Cross-Attention and Temporally-Adaptive RoPE modules provide the structural basis to process varying subject counts and independent action schedules concurrently.
> Specifically, in **Figure 6 (left)** of the revised manuscript, two subjects follow separate action schedules: the *Suited Man* performs "Adjust the tie" ($0–3s$) followed by "Absorbed in the phone" ($3–5s$), while the *Woman* executes "Lower head" ($0–1.5s$), "Gaze ahead" ($1.5–3.5s$), and "Check watch" ($3.5–5s$). This capability is further highlighted in **Figure 6 (right)**, where temporal independence is even more pronounced. From $0–3.5s$, the *Man in an apron* is instructed to "Read a newspaper" while the *Girl* performs "Look right and hold the teddy bear." Concurrently, the *Woman* and *Old man* engage in a conversational action sequence ("Look right" followed by "Talk"). The generated videos faithfully reflect these overlapping and asynchronous actions, confirming the model's robustness in handling complex multi-subject temporal dynamics.
>
> **[W4] Customizing animals and general objects.**
>
> Yes, ReactID supports the customization of non-human entities, including animals and general objects, as demonstrated in **Figure 5 (left)** and **Figure 6** of the revised manuscript.
> This capability is underpinned by our comprehensive taxonomy, constructed by aggregating labels from multiple datasets to cover humans, animals, and general objects. This diversity enables ReactID to generalize effectively across a broad spectrum of categories.
> In **Figure 5 (left)** of the revised manuscript, we showcase a generation example involving the customization of a dog and a teddy bear. The generated video faithfully depicts the dog interacting with the teddy bear in strict adherence to the prompt. Additionally, **Figure 6 (left)** demonstrates the successful integration of a plant as a consistent subject alongside human actors. Similarly, in **Figure 6 (right)**, the teddy bear held in the girl's arms maintains a consistent appearance aligned with the reference image throughout the sequence. Collectively, these qualitative results validate ReactID's capability to generalize personalization to diverse non-human categories while maintaining high identity fidelity.

---

> ### Author Response · Authors · 2025-11-23
> **Response to Reviewer PVcC (2/2)**
>
> **[W5] More comprehensive references.**
>
> Thanks for your valuable suggestions. As suggested, we have added the discussion of all the mentioned references in the revised Section 2 (Related Work).
> CustomCrafter fine-tunes self-attention layers to enable subject customization without compromising the inherent motion priors of UNet, thereby ensuring both high identity fidelity and natural video dynamics. Moreover, customization learning are not limited to visual appearance and identity, but also extended to relations and interactions. Pioneering this direction, ReVersion proposes a diffusion-based relation inversion framework to capture specific relation from images, which is further employed for image generation. DreamRelation further extends such type of relation modeling into video synthesis.

---

### Author Response · Authors · 2025-12-03
**Rebuttal Summary**

We sincerely thank the AC and all reviewers for their constructive feedback and recognition of our work. We are encouraged that reviewers appreciated the **clear motivation** tackling the identity-action trade-off (*all*), the **novel framework** integrating data, training, and timeline conditioning (*all*), the meticulously designed **data pipeline and dataset** (*all*), the **solid experiments** demonstrating state-of-the-art performance (*PVcC, 3tVk, GCZR*), and **clear presentation** (*PVcC, GCZR*).

We have thoroughly addressed the reviewers' concerns and incorporated extensive improvements in the revision. Below is a summary of our responses to the major concerns and the key enhancements made.

### Response to Major Concerns
**Concern #1 by Reviewer vCvz (Rating 4 / Confidence 4): Quality of Timeline Annotations in ReactID-Data**

**Response:** Reviewer vCvz raised concerns regarding the quality of timeline annotations. In response, we conducted a human study (Table A), verifying that our ensemble pipeline achieves a high F1 score of **0.78**, significantly outperforming top-tier automated baselines (best competitor at **0.71**). These results confirm that our approach delivers superior annotation quality, offering a robust solution that effectively eliminates the need for labor-intensive manual annotation for massive video datasets.


**Concern #2 by Reviewer 3tVk (Rating 4 / Confidence 1): Impact of Potential Errors in LLM-Planned Timeline Conversion**

**Response:** Reviewer 3tVk questioned whether LLM-planned timelines introduce errors during inference. To address this, we performed a human preference comparison and a blind test between LLM-planned and Human Expert-planned timelines. The confusion matrix (Table F) reveals a classification accuracy near 50% (indicating chance-level distinguishability), confirming that LLM-generated timelines are perceptually indistinguishable from human expert annotations. Furthermore, user preference studies (Table E) show no performance degradation, verifying that the LLM planner is robust and not a bottleneck.

### Improvements in Revision
We also carefully addressed all other feedback and incorporated extensive improvements as suggested to further strengthen the manuscript:

**1. More visualizations demonstrating generalization capabilities:** We added additional visualizations and qualitative comparisons to illustrate ReactID's ability to generalize to challenging settings, including:
- **three or more subjects** (Figure 5, Figure 6, Appendix Figure 13);
- **independent asynchronous action timelines** (Figure 6);
- **animals & objects customization** (Figure 5, Figure 6, Appendix Figure 13);
- **long video generation** (Appendix Figure 11).

**2. More evaluations on mask annotations and module designs:** We conducted detailed analyses on subject mask annotations (Table B), and decoupled the contribution of our high-quality data and timeline-aware model design via further ablation studies (Table G).

**3. More discussion on efficiency/scalability and detailed curriculum learning:** We provided a detailed explanation and discussion regarding training & inference efficiency, scalability, curriculum details to ensure reproducibility and improve clarity.

### Conclusion
Overall, we are glad that the reviewers found our method innovative and well-presented. We have thoroughly addressed all concerns and diligently incorporated suggested improvements in the revised manuscript to further substantiate our contributions. We believe these revisions solidify the quality and rigor of our work. We once again thank the reviewers and AC for their efforts, and we hope this summary facilitates a comprehensive assessment of our work.

Sincerely,
Authors

---

### Meta-Review · Area_Chair_nJ2D · 2026-01-13

**Summary:**

The reviewers identified several key concerns that needed to be addressed for this paper on personalized video generation. The main concerns centered around:
- Dataset/code release and reproducibility
- Generalization to multi-subject scenarios (3+ subjects) and complex independent action timelines.
- Quality of timeline annotations and subject mask annotations.
- Training/inference efficiency compared to baselines.
- Impact of LLM errors in timeline conversion.
- Ablation studies on data annotation quality.
- Support for non-human subjects (animals and objects)
- Long video generation capabilities.

The authors generally addressed these concerns in their rebuttal with new experiments, human studies, detailed specifications, and additional figures showing results for challenging cases.

**Reviewer Concerns:**

Addressed Concerns:
- Reviewer PVcC:
  - Confirmed open-source commitment with detailed plans
  - Multi-subject generalization (Added new figures Figure 5, Figure 6) showing 3+ subjects
  - Complex action timelines: Demonstrated in Figure 6 with multiple subjects performing distinct actions
  - Animals/objects: Added examples with dogs, teddy bears, and plants in Figures 5 and 6
  - References: Added discussion of CustomCrafter in Related Work

- Reviewer vCvz:
  - Timeline annotation quality: Added human study showing F1 score of 0.78 (vs. baselines at 0.71)
  - Mask annotation robustness: Added human preference study showing 54% preference rate (vs. 31% for Vanilla SAM)
  - Curriculum details: Provided specific schedules for difficulty threshold, mixing ratio, and weights

- Reviewer 3tVk:
  - Training/inference efficiency: Added Table D showing only 5.3% higher training time and 2.9% higher inference time
  - LLM error impact: Added human preference study showing negligible perceptual difference
  - 3+ subjects: Added Figure 6 showing a family of 4 subjects with distinct actions
  - Long video generation: Added 30-second examples in Appendix Figure 11

- Reviewer GCZR:
  - Ablation on data annotation quality: Added Table G showing a detailed ablation study on the impact of high-precision data and timeline annotations.

Outstanding Concerns:
- Although the authors demonstrate generalization to non-human subjects, multi-entity scenarios (≥3 entities) and long video generation through qualitative examples, quantitative metrics evaluated under the same protocol as the main paper would strengthen the claim. Examples are weak to demonstrate the capability.

**Reviewer Scores:**

- PVcC (original 6): Would likely be at least 6. The authors fully addressed all concerns with evidence and new results.
- vCvz (original 4): Would likely increase to 6. The rebuttal provided comprehensive evidence for the quality of timeline annotations and mask robustness, and detailed curriculum specifications.
- 3tVk (original 4): Would likely be at least 4. The rebuttal solve some problems but the issue on generalization is only solved by examples rather than rigorous quantitative evaluation.
- GCZR (original 6): Would likely keep at 6. The additional ablation study strengthened their contribution.

The paper addresses a challenge in personalized video generation with a well-motivated, comprehensive solution. The authors' rebuttal was thorough, providing necessary evidence for reviewer concerns. Although the generalization issue is not comprehensively addressed, the work makes a meaningful contribution to the field with datasets and method contributions. Therefore, I lean towards acceptance.

---

### Decision · Program_Chairs · 2026-01-26

Accept (Poster)